# GLIMPSE: ENABLING WHITE-BOX METHODS TO USE PROPRIETARY MODELS FOR ZERO-SHOT LLM-GENERATED TEXT DETECTION

**Guangsheng Bao [1,2], Yanbin Zhao [3], Juncai He [4], Yue Zhang [2,*]**

[1] Zhejiang University      [2] School of Engineering, Westlake University
[3] School of Mathematics, Physics and Statistics, Shanghai Polytechnic University
[4] Computer, Electrical and Mathematical Science and Engineering Division,
King Abdullah University of Science and Technology
`zhaoyb553@nenu.edu.cn`     `juncai.he@kaust.edu.sa`
`{baoguangsheng, zhangyue}@westlake.edu.cn`

## ABSTRACT

Advanced large language models (LLMs) can generate text almost indistinguishable from human-written text, highlighting the importance of LLM-generated text detection. However, current zero-shot techniques face challenges as white-box methods are restricted to use weaker open-source LLMs, and black-box methods are limited by partial observation from stronger proprietary LLMs. It seems impossible to enable white-box methods to use proprietary models because API-level access to the models neither provides full predictive distributions nor inner embeddings. To traverse the divide, we propose *Glimpse*, a probability distribution estimation approach, predicting the full distributions from partial observations.[1] Despite the simplicity of Glimpse, we successfully extend white-box methods like Entropy, Rank, Log-Rank, and Fast-DetectGPT to latest proprietary models. Experiments show that Glimpse with Fast-DetectGPT and GPT-3.5 achieves an average AUROC of about 0.95 in five latest source models, improving the score by 51% relative to the remaining space of the open source baseline. It demonstrates that the latest LLMs can effectively detect their own outputs, suggesting that advanced LLMs may be the best shield against themselves.

## 1 INTRODUCTION

Large language models (LLMs) (OpenAI, 2022; Team et al., 2023; Anthropic, 2024) can produce fluent and coherent text content, which is almost indistinguishable from human-written content (Ippolito et al., 2020; Shahid et al., 2022; Dugan et al., 2023). It powers the productivity of various industries such as journalism (Christian, 2023), social media (Yuan et al., 2022), and education (M Alshater, 2022), but at the same time it causes various risks such as misinformation, disinformation, and plagiarism (Pan et al., 2023; Weidinger et al., 2021; Meyer et al., 2023), thus urging automatic detection tools for building trustworthy AI systems (Kaur et al., 2022; Sun et al., 2024). However, as long as LLMs increase their ability, detection of their generations becomes more difficult (Mireshghallah et al., 2023; Sadasivan et al., 2023; Tang et al., 2024).

The most powerful LLMs are generally proprietary models, which only provide limit access through API. Consequently, the existing white-box methods that require 'full access' cannot be applied to these models, and instead, various black-box methods are developed. Black-box methods, such as DetectGPT (Mitchell et al., 2023) and DNA-GPT (Yang et al., 2023a), demonstrate competitive detection accuracies in advanced LLMs such as GPT-3, ChatGPT, and GPT-4. However, these methods are less efficient and less robust compared to their white-box counterparts (Bao et al., 2023), because they rely on knowing the source model and need multiple evaluations or generations

---

[*]Corresponding author.
[1]We release our code and data at `https://github.com/baoguangsheng/glimpse`.

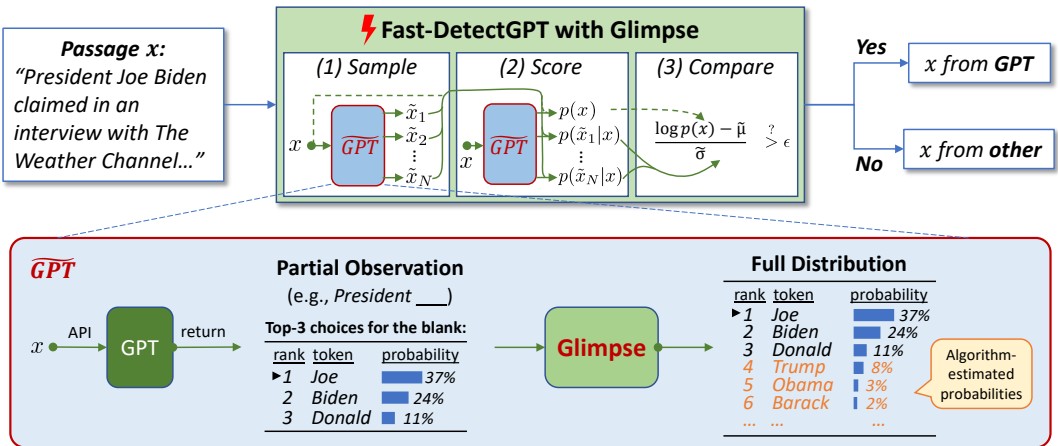

Figure 1: Take Fast-DetectGPT as an example to apply *Glimpse*. The notion $\widetilde{\text{GPT}}$ refers to the model with estimated distribution, where the partial observation (top-$K$ probabilities) returned by the model API is completed into a full distribution. The '*token*' column is just for reference, which is not necessary for calculating the metric (conditional probability curvature).

of text sequences. Instead of improving these black-box methods like Su et al. (2023), we turn back to white-box methods to see the potential of combining their strength with the power of the latest LLMs. [2]

For example, Fast-DetectGPT (Bao et al., 2023) uses a fixed surrogate model to detect text from various source models, including ChatGPT and GPT-4. The basic idea is to use a surrogate model to obtain token distributions to calculate a metric of conditional probability curvature, where the higher the metric, the more likely the input is machine-generated. Despite its simplicity, it achieves detection accuracies that are higher than those of the black-box methods. However, its detection accuracy on GPT-4 (AUROC 0.91) is significantly lower than on open-source models (an average of 0.99 on five models). We speculate that the lower accuracies in the latest models are caused by the distribution mismatch between the small surrogate model and the large source models, which can potentially be addressed by using the latest large models as the surrogate.

To enable white-box methods to use proprietary models, we propose *Glimpse*, a probability distribution estimation approach, to estimate the full distribution from the partial observation returned by the model API. This observation includes the probabilities (logprobs) of the input tokens and a few (at least 1) top-ranking tokens on each token position. [3] Specifically, take Fast-DetectGPT as an example, illustrated in Figure 1. We first obtain the top probabilities from the GPT model and then use these probabilities to estimate the distribution across the entire vocabulary. The basic idea is to find an empirical correlation between the top probabilities and the full vocabulary probabilities. To this end, we consider the parameterized Geometric distribution, Zipfian distribution, and an MLP model trained on the data to model the correlation. Using Glimpse, we also extend methods like Entropy, Rank, and LogRank to proprietary models.

Experiments show exceptionally strong results for these white-box methods using the latest LLM, where Rank, LogRank, and Fast-DetectGPT using Glimpse outperform their versions using open-source LLMs with significant margins (more than 25% relative to the remaining space). We achieve the best detection accuracies for five source models, obtaining an accuracy of 0.98 for ChatGPT, 0.94 for GPT-4, 0.96 for Claude-3 Sonnet, 0.97 for Claude-3 Opus, and 0.92 for Gemini-1.5 Pro in the stimulated white-box setting. These results indicate that the latest LLMs can detect their generations with high accuracy.

To our knowledge, we are the first to investigate white-box detection methods using proprietary models, achieving significantly improved detection accuracies among zero-shot methods, demonstrating that *the most powerful LLMs may be the strongest 'shield' against themselves*.

---

[2] By 'latest', we refer to the most powerful models that are widely used like ChatGPT and GPT-4.

[3] Provided by Completion API of popular models like Google's PaLM-2 (Anil et al., 2023), OpenAI's GPT-3 (Brown et al., 2020), GPT-3.5 (OpenAI, 2022), and GPT-4 (OpenAI, 2023).

## 2 METHOD

We use Fast-DetectGPT (Bao et al., 2023) as an example to illustrate how to apply Glimpse to existing white-box methods in Section 2.2, and present three specific probability distribution estimation algorithms in Section 2.3. We further discuss three more white-box methods with Glimpse in Section 2.4.

### 2.1 TASK AND SETTINGS

We are focusing on the zero-shot detection of LLM-generated text, which we frame as a binary classification problem: determining whether a given text was created by a model or a human. A zero-shot detector usually uses a scoring model to produce a detection metric and makes a decision by comparing this metric to a predetermined threshold. The scoring model can either be the same as the source model or a different one, for example a fixed delegate model. The access level to this scoring model can vary, and methods can be categorized into black-box or white-box methods based on this (similar to Tang et al. (2024)).

**Black-box methods** use API-level access to interact with the scoring model, for example proprietary GPT-3.5, while **white-box methods** assume full access to the scoring model, for example open source Neo-2.7B. In this definition, methods such as Fast-DetectGPT (Bao et al., 2023) and PHD (Tulchinskii et al., 2024), which employ a delegated open source model to identify generations from proprietary LLMs, are classified as white-box methods. It is crucial to note that this definition differs from Yang et al. (2023b), which defines black box and white box based on the access level to the source model.

### 2.2 FAST-DETECTGPT WITH GLIMPSE

Adding Glimpse does not change the overall framework of the baseline. It only replaces the model distribution $p_\theta$ with the estimated distribution $\tilde{p}_\theta$ when the missing part of $p_\theta$ is required. In the following description, we rehearsal the framework of Fast-DetectGPT with the updated distribution.

Fast-DetectGPT posits that human and machine language generation differs in word selection based on context. While machines favor words with higher model probabilities, humans do not necessarily demonstrate such tendency. This discrepancy is quantified using a metric, naming the conditional probability curvature. And we decide if a text is machine generated by comparing the metric with a threshold $\epsilon$, which is chosen according to specific scenarios.

Formally, given a text passage $x$, a proprietary model $p_\theta$, and an estimated distribution $\tilde{p}_\theta$, the *conditional probability function* defined by Fast-DetectGPT is expressed as

$$p_\theta(\tilde{x}|x) = \prod_j p_\theta(\tilde{x}_j|x_{<j}),\tag{1}$$

which denotes the predictive distribution of the model taking $x$ as the input. As a special case, when $\tilde{x}$ equals to $x$, $p_\theta(x|x) = p_\theta(x)$. Take the second word position ($j = 2$) of the passage in Figure 1 as an example. Its context is $x_{<2} =$ *'President'*, and the probabilities for possible choices $\tilde{x}_2 \in [$*'Joe'*, *'Biden'*, *'Donald'*, ...$]$ are $[0.37, 0.24, 0.11, ...]$. The tokens $\tilde{x}_j$ for different $j$ are independent of each other given the input. This conditional independence allows for an efficient calculation in the algorithm.

Using the conditional probability function, the *conditional probability curvature* defined by Fast-DetectGPT is written as

$$\mathbf{d}(x, p_\theta) = \frac{\log p_\theta(x) - \tilde{\mu}}{\tilde{\sigma}},\tag{2}$$

where the token likelihoods $p_\theta(x_j|x_{<j})$ are provided directly by the proprietary model, like $p_\theta(x_2 =$ *'Joe'*$|x_{<2} =$ *'President'*$) = 0.37$, no matter whether $x_j$ is among the top-$K$ choices. The expected

score $\tilde{\mu}$ and the expected variance $\tilde{\sigma}^2$ are computed using the estimated distribution $\tilde{p}_\theta$ as

$$\tilde{\mu} = \mathbb{E}_{\tilde{x}\sim\tilde{p}_\theta(\tilde{x}|x)}\left[\log\tilde{p}_\theta(\tilde{x}|x)\right] = \sum_j \mathbb{E}_{\tilde{x}_j\sim\tilde{p}_\theta(\tilde{x}_j|x_{<j})}\left[\log\tilde{p}_\theta(\tilde{x}_j|x_{<j})\right] = \sum_j \tilde{\mu}_j,$$

$$\tilde{\sigma}^2 = \mathbb{E}_{\tilde{x}\sim\tilde{p}_\theta(\tilde{x}|x)}\left[(\log\tilde{p}_\theta(\tilde{x}|x) - \tilde{\mu})^2\right] = \sum_j \mathbb{E}_{\tilde{x}_j\sim\tilde{p}_\theta(\tilde{x}_j|x_{<j})}\left[(\log\tilde{p}_\theta(\tilde{x}_j|x_{<j}) - \tilde{\mu}_j)^2\right] = \sum_j \tilde{\sigma}_j^2.$$

$$(3)$$

In the expectations, $\tilde{x} \sim \tilde{p}_\theta(\tilde{x}|x)$ denotes the sampling step, while $\log\tilde{p}_\theta(\tilde{x}|x)$ denotes the scoring step, as illustrated in Figure 1. The expectations over the whole sequence are calculated by accumulating all token-level expectations.

The token-level mean $\tilde{\mu}_j$ signifies the entropy of the predictive distribution at the j-th token. We calculate it analytically by enumerating all possible choices

$$\tilde{\mu}_j = \mathbb{E}_{\tilde{x}_j\sim\tilde{p}_\theta(\tilde{x}_j|x_{<j})}\left[\log\tilde{p}_\theta(\tilde{x}_j|x_{<j})\right] = \sum_{\tilde{x}_j}\tilde{p}_\theta(\tilde{x}_j|x_{<j})\log\tilde{p}_\theta(\tilde{x}_j|x_{<j}),\qquad(4)$$

such as $\tilde{\mu}_2 = 0.37 \cdot \log 0.37 + 0.24 \cdot \log 0.24 + 0.11 \cdot \log 0.11 + ....$

The token-level variance $\tilde{\sigma}_j^2$ is also computed analytically as

$$\tilde{\sigma}_j^2 = \mathbb{E}_{\tilde{x}_j\sim\tilde{p}_\theta(\tilde{x}_j|x_{<j})}\left[(\log\tilde{p}_\theta(\tilde{x}_j|x_{<j}) - \tilde{\mu}_j)^2\right] = \sum_{\tilde{x}_j}\tilde{p}_\theta(\tilde{x}_j|x_{<j})\log^2\tilde{p}_\theta(\tilde{x}_j|x_{<j}) - \tilde{\mu}_j^2,\quad(5)$$

where the summation over $\tilde{x}_j$ is similarly achieved by enumerating all possible choices, like $\tilde{\sigma}_2^2 = 0.37 \cdot \log^2 0.37 + 0.24 \cdot \log^2 0.24 + ... - \tilde{\mu}_2^2.$

### 2.3 GLIMPSE: A PROBABILITY DISTRIBUTION ESTIMATION APPROACH

Formally, given a text sequence $x$, the proprietary model $p_\theta(\tilde{x}|x)$ provides us the likelihood $p_\theta(x_j|x_{<j})$ and the top-$K$ token probabilities $p_\theta(\tilde{x}_j^k|x_{<j})|_{k=1}^K$ on each token position $j$, where $k$ denotes the rank. The problem is then formulated as estimating $p_\theta(\tilde{x}_j|x_{<j})$ over the whole vocabulary according to the given information.

However, in general, we do not need token-probability pairs for metric calculation. Take Fast-DetectGPT as an example. The mean $\tilde{\mu}_j$ and variance $\tilde{\sigma}_j^2$ only depend on the probability values in the distribution. That is to say, we can calculate them using only the '*probability*' column in the full distribution in Figure 1, where the '*token*' column is not necessary. This advantage is not specific to Fast-DetectGPT. Other methods like Entropy, Rank, and Log-Rank can also be calculated from the probabilities only.

To simplify the discussion in the following, we denote the probabilities as $p(k)$, omitting the expression of the token $\tilde{x}_j^k$ and position $j$. Using the top-$K$ ($K = 3$ typically) probabilities $p(k)|_{k=1}^K$, we estimate the rest $p(k)|_{k=K+1}^M$, where $M$ denotes the size of the list. It is worth noting that $M$ is not necessarily as large as the vocabulary size, because large ranks generally correspond to low probabilities. When the probabilities are small enough, their effects on the metric are ignorable.

Consequently, $\tilde{\mu}_j$ and $\tilde{\sigma}_j^2$ are rewritten as

$$\tilde{\mu}_j = \sum_{\tilde{x}_j}\tilde{p}_\theta(\tilde{x}_j|x_{<j})\log\tilde{p}_\theta(\tilde{x}_j|x_{<j}) = \sum_k p(k)\log p(k),$$

$$\tilde{\sigma}_j^2 = \sum_{\tilde{x}_j}\tilde{p}_\theta(\tilde{x}_j|x_{<j})\log^2\tilde{p}_\theta(\tilde{x}_j|x_{<j}) - \tilde{\mu}_j^2 = \sum_k p(k)\log^2 p(k) - \tilde{\mu}_j^2,$$

$$(6)$$

where the summation over all possible tokens $\tilde{x}_j$ is thus converted into the summation over all possible ranks $k$.

The probability distribution across ranks generally follows a decaying pattern, where the larger models tend to have a higher top-1 probability and a bigger decay factor demonstrating a sharper distribution (Appendix A). We approximate the pattern using parameterized distributions, allocating the remaining probability mass (excluding top-$K$ probabilities) to ranks larger than $K$.

Each parameterized distribution represents a family of distributions controlled by some parameters. Traditionally, we fit the distribution to the data points to estimate the parameter, where we could use the top-$K$ probabilities as the data points. However, this approach may underutilize the observed information and infer top-$K$ probabilities different from the real ones.

In this study, we consider the estimation problem in constraints. The basic constraints include: 1) *total probability constraint* – a summation of the probabilities equals 1. 2) *monotonic decrease constraint* – the probability of a higher rank is lower. Using the total probability constraint, we decide the probability mass to allocate to ranks larger than K. Using the monotone decrease constraint, we decide on a suitable family of distributions. We discuss three specific estimation algorithms as follows.

**Estimation Using Geometric Distribution.** As the simplest decaying pattern, we consider exponential decay with a fixed decay factor, resulting in a Geometric distribution. We extend the distribution to multiple top probabilities and a limited range of $k$. We express the probabilities for ranks larger than $K$ in near-Geometric distribution that

$$\begin{cases} p(k) = p_k, & \text{for } k \in [1..K] \\ p(k) = p_K \cdot \lambda^{k-K}, & \text{for } k \in [K+1..M] \\ \sum_{k=1}^{M} p(k) = 1, \end{cases} \tag{7}$$

where $\lambda$ is a decay factor in $(0, 1)$, and $M$ is the size of the rank list. We solve $\lambda$ under the constraints as detailed in Appendix A.1.

**Estimation Using Zipfian Distribution.** Frequencies of words in natural languages usually adhere to Zipf's law (Zipf, 1946; 2013), where the word frequency and word rank follow a Zipfian distribution. Assuming that the word frequencies in a given context also comply with this law, we consider it as an alternative distribution for our estimation. Given the top-$K$ probabilities $p_k$, we compute the probabilities of tokens with a ranking greater than $K$ in a Zipfian distribution

$$\begin{cases} p(k) = p_k, & \text{for } k \in [1..K] \\ p(k) = p_K \cdot (\frac{\beta}{k-K+\beta})^{\alpha}, & \text{for } k \in [K+1..M] \\ \sum_{k=1}^{M} p(k) = 1, \end{cases} \tag{8}$$

where $\alpha$ and $\beta$ are two positive parameters. $M$ is the size of the rank list. To solve the two parameters, we introduce an additional loss function to minimize their deviations from typical values. We identify the best $\alpha$ and $\beta$ by searching a loss table $\text{Loss}(\alpha, \beta)$ as detailed in Appendix A.2.

**Estimation Using a MLP Model.** The Geometric and Zipfian algorithms both work on assumptions about the distributions. An alternate approach that does not rely on these assumptions involves modeling the distribution within a neural network. We consider the simple Multi-Layer Perceptron (MLP) model with a single hidden layer, which accepts the top-$K$ probabilities and predicts the probabilities for the rest of the ranks. The distribution is expressed as

$$\begin{cases} p(k) = p_k, & \text{for } k \in [1..K] \\ p(k) = p_{\text{rest}} \cdot p_{\text{MLP}_\theta}(k - K), & \text{for } k \in [K+1..M] \\ \sum_{k=1}^{M} p(k) = 1, \end{cases} \tag{9}$$

where $p_{rest} = 1 - \sum_{k=1}^{K} p_k$ and $p_{\text{MLP}_\theta}$ represents the MLP predictive distribution. The MLP is defined in detail, including training and inference, in Appendix A.3.

## 2.4 UNIVERSALITY OF GLIMPSE

Glimpse can also be used by other zero-shot detection methods like Entropy, Rank, and LogRank. For Entropy, the calculation is straightforward by summing $p(k) \cdot \log p(k)$ over all items in the rank list. For Rank and LogRank, we decide the rank of the current token by searching the closest $p(k)$ compared to the probability of the current token $p_\theta(x_j|x_{<j})$. In this study, we only examine the basic white-box methods, leaving the integration of more sophisticated white-box methods and additional estimation algorithms to future research.

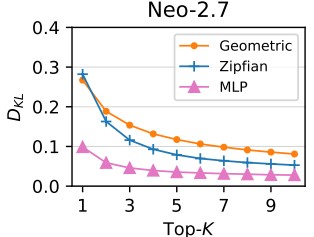
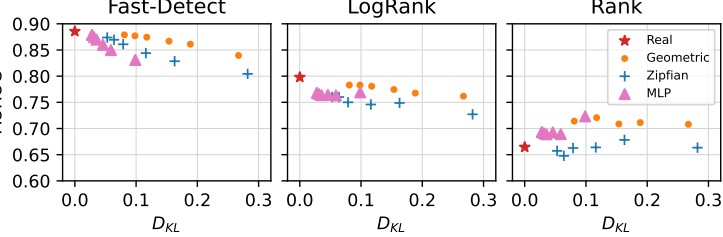

Figure 2: *KL divergence against real distributions from Neo-2.7B.*

Figure 3: Correlation between *AUROC* and *KL divergence*, evaluated on XSum produced by GPT-4. We use the open-source model Neo-2.7B as the scoring model for Glimpse algorithms.

# 3 EXPERIMENTS

## 3.1 SETTINGS

We evaluate our methods on four datasets that cover seven languages, five source models that cover three model families, and four scoring models from small to large. We run each main experiment three times and report the median. The metrics used are described in Appendix B.1.

**Datasets.** We follow studies (Mitchell et al., 2023; Bao et al., 2023; Yang et al., 2023a; Zeng et al., 2024a) on the evaluation datasets and their settings: *XSum* for news (Narayan et al., 2018), *Writing* for story (Fan et al., 2018), and *PubMed* for technical question answer (Jin et al., 2019) , where for each dataset 150 human-written samples are randomly selected and corresponding LLM texts are generated using the same prefix (30 tokens for articles or questions for QAs). Furthermore, we use *M4* (Wang et al., 2024) to incorporate various languages such as *Chinese*, *Russian*, *Urdu*, *Indonesian*, *Arabic*, and *Bulgarian* , where for each language 150 pairs are randomly selected from the ChatGPT subsets. To evaluate the detectors ability to handle diverse domains and languages, we combine XSum, Writing, and PubMed into a single dataset called *Mix3*, and the six language datasets into another called *Mix6*.

**Source Models.** We evaluate our detector on five latest LLMs from different companies, including *ChatGPT* (gpt-3.5-turbo) (OpenAI, 2022), *GPT-4* (gpt-4) (OpenAI, 2023), *Claude3 Sonnet* (claude-3-sonnet-20240229) and *Opus* (claude-3-opus-20240229) (Anthropic, 2024), *Gemini-1.5 Pro* (gemini-1.5-pro) (Team et al., 2023), where Opus is supposed at the same level as GPT-4 and Sonnet at the same level as ChatGPT. We use the ChatCompletion API [4] of these models to prepare the datasets.

**Scoring Models.** A good scoring model can detect generations from a wide range of source models. We use OpenAI GPT series (from small to large) as the scoring model, inlcuding *Babbage* (babbage-002, 1.3B) and *Davinci* (davinci-002, 175B) (Brown et al., 2020), *GPT-3.5* (gpt-35-turbo-0301 or gpt-35-turbo-1106 almost equally, 175B) (OpenAI, 2022), and *GPT-4* (gpt-4-1106) (OpenAI, 2023) as scoring models using AzureOpenAI (see Appendix B.2). For comparison, we use *Neo-2.7* (gpt-neo-2.7B) (Black et al., 2021), *Phi2-2.7B* (Javaheripi et al., 2023), *Qwen2.5-7B* (Yang et al., 2024; Team, 2024), and *Llama3-8B* (Dubey et al., 2024) as representatives of open-source models, run locally on a Tesla A100 GPU.

**Baselines.** Among zero-shot detectors, we compare our methods with existing solutions such as *Fast-DetectGPT* (shortly Fast-Detect) (Bao et al., 2023), *DetectGPT* (Mitchell et al., 2023), *DNA-GPT* (Yang et al., 2023a), and simple baselines such as *Likelihood* (equivalent to perplexity), *Entropy*, *Rank*, and *Log-Rank* (Gehrmann et al., 2019; Solaiman et al., 2019; Ippolito et al., 2020). For other detectors, we compare our methods to commercial *GPTZero* (Tian & Cui, 2023).

**Hyper-Parameters.** By default, we use top-5 probabilities, a rank list size of 1000 for Geometric and 100 for Zipfian and MLP, prompt4 from Table 6 for GPT-4 and prompt3 for other scoring models. These settings are ablated in Section 3.4 with one hyper-parameter changed each time, where we report the average accuracy over XSum, Writing, and PubMed.

---

[4] https://platform.openai.com/docs/guides/text-generation/chat-completions-api

Table 1: *Main results for Glimpse*, where the columns '*GPT-4*', '*Claude-3*', and '*Gemini-1.5*' display the AUROCs on the diverse Mix3 dataset, with the detailed results for GPT-4 and Gemini-1.5 in Table 3 and Claude-3 in Table 4 in Appendix C. We mark the best in each column in **bold**. Methods marked with ◇ require multiple evaluations or generations to detect one passage, thereby, consuming multiple times of cost and time. Additionally, we make a comparison of ACC in Table 5.

| Method | | ChatGPT | | ChatGPT | GPT-4 | Claude-3 | | Gemini-1.5 | Avg. |
|---|---|---|---|---|---|---|---|---|---|
| | XSum | Writing | PubMed | Mix3 | Mix3 | Sonnet | Opus/Mix3 | Pro/Mix3 | Mix3 |
| GPTZero | 0.9843 | 0.9303 | 0.8403 | 0.9143 | 0.9009 | - | - | - | - |
| **Zero-Shot Detectors Using White-Box Open-Source LLMs** | | | | | | | | | |
| Likelihood (Neo-2.7) | 0.9578 | 0.9740 | 0.8775 | 0.9071 | 0.7690 | 0.8661 | 0.9030 | 0.7416 | 0.8373 |
| Entropy (Neo-2.7) | 0.3305 | 0.1902 | 0.2767 | 0.3136 | 0.4114 | 0.3466 | 0.3265 | 0.3959 | 0.3588 |
| Rank (Neo-2.7) | 0.7494 | 0.8064 | 0.5979 | 0.7044 | 0.6448 | 0.6888 | 0.7056 | 0.6260 | 0.6739 |
| LogRank (Neo-2.7) | 0.9582 | 0.9656 | 0.8687 | 0.9059 | 0.7626 | 0.8654 | 0.9042 | 0.7353 | 0.8347 |
| DNA-GPT (Neo-2.7) ◇ | 0.9124 | 0.9425 | 0.7959 | 0.7192 | 0.6430 | 0.7080 | 0.7326 | 0.6438 | 0.6893 |
| DetectGPT (T5-11B/Neo-2.7) ◇ | 0.8416 | 0.8811 | 0.7444 | 0.7826 | 0.6136 | 0.7967 | 0.7776 | 0.7406 | 0.7422 |
| Fast-Detect (GPT-J/Neo-2.7) | 0.9907 | 0.9916 | 0.9021 | 0.9487 | 0.8999 | 0.9260 | 0.9468 | 0.8072 | 0.9057 |
| Fast-Detect (Phi2-2.7B) | 0.8096 | 0.7245 | 0.8121 | 0.7627 | 0.5742 | 0.6957 | 0.7450 | 0.6164 | 0.6788 |
| Fast-Detect (Qwen2.5-7B) | 0.7808 | 0.8117 | 0.7887 | 0.7655 | 0.6862 | 0.7813 | 0.8119 | 0.6839 | 0.7458 |
| Fast-Detect (Llama3-8B) | 0.8508 | 0.8446 | 0.7941 | 0.7796 | 0.7269 | 0.8212 | 0.8510 | 0.7552 | 0.7868 |
| **Zero-Shot Detectors Using Black-Box Proprietary LLMs** | | | | | | | | | |
| Likelihood (GPT-3.5) | 0.9203 | 0.9925 | 0.9544 | 0.9246 | 0.8029 | 0.9023 | 0.9295 | 0.8043 | 0.8727 |
| DNA-GPT (GPT-3.5) ◇ | 0.9260 | 0.9329 | 0.9304 | 0.8369 | 0.7748 | 0.7871 | 0.8383 | 0.7107 | 0.7896 |
| *Glimpse using Geometric* | | | | | | | | | |
| Entropy (GPT-3.5) | 0.3188 | 0.0463 | 0.1793 | 0.2160 | 0.4074 | 0.2582 | 0.2339 | 0.4144 | 0.3060 |
| Rank (GPT-3.5) | 0.8577 | 0.9845 | 0.8383 | 0.8533 | 0.7395 | 0.8473 | 0.8645 | 0.7406 | 0.8090 |
| LogRank (GPT-3.5) | 0.9240 | 0.9931 | 0.9532 | 0.9277 | 0.7870 | 0.9062 | 0.9336 | 0.7872 | 0.8683 |
| Fast-Detect (Babbage) | **0.9908** | 0.9904 | 0.9570 | 0.9764 | 0.8974 | 0.9438 | 0.9698 | 0.8083 | 0.9191 |
| Fast-Detect (Davinci) | 0.9900 | **0.9976** | 0.9421 | 0.9763 | 0.9131 | 0.9606 | 0.9742 | 0.8601 | 0.9369 |
| Fast-Detect (GPT-3.5) | 0.9808 | 0.9972 | **0.9702** | 0.9766 | 0.9411 | 0.9576 | 0.9689 | **0.9244** | 0.9537 |
| Fast-Detect (GPT-4) | 0.9815 | 0.9935 | 0.9564 | 0.9735 | 0.9647 | 0.9623 | **0.9817** | 0.8947 | 0.9554 |
| *Glimpse using Zipfian* | | | | | | | | | |
| Fast-Detect (GPT-3.5) | 0.9826 | 0.9956 | 0.9639 | 0.9647 | 0.9319 | 0.9475 | 0.9588 | 0.9161 | 0.9438 |
| Fast-Detect (GPT-4) | 0.9885 | 0.9917 | 0.9461 | 0.9768 | **0.9719** | 0.9613 | 0.9792 | 0.8991 | 0.9576 |
| *Glimpse using MLP* | | | | | | | | | |
| Fast-Detect (GPT-3.5) | 0.9819 | 0.9959 | 0.9676 | 0.9702 | 0.9342 | 0.9526 | 0.9634 | 0.9184 | 0.9478 |
| Fast-Detect (GPT-4) | 0.9869 | 0.9930 | 0.9528 | **0.9771** | 0.9705 | **0.9631** | 0.9807 | 0.9001 | **0.9583** |

## 3.2 THE EFFECTIVENESS OF GLIMPSE

We assess the effectiveness of Glimpse by comparing the estimated distributions with the real distributions, using Neo-2.7B as the scoring model. As Figure 2 shows, their Kullback–Leibler (KL) divergence decreases as long as more top probabilities are used. Overall, MLP obtains the lowest divergence, while Geometric distribution has the highest divergence, suggesting the most accurate estimation of MLP. However, the correlation between KL divergences and detection accuracies varies for different detection methods and estimation algorithms. As Figure 3 illustrates, the AUROC *declines* when the divergence increases for Glimpse (Fast-Detect and LogRank) with any of the estimation algorithms, while the AUROC *inclines* for Glimpse (Rank). Although the geometric distribution has larger divergences in general, it achieves higher or equal detection accuracies than other algorithms, suggesting that the effect of Glimpse is beyond the similarity of estimated distributions.

It is worth noting that the accuracies achieved by Glimpse are not significantly lower than the baseline using real distribution for Fast-Detect and LogRank, and are even significantly higher than the baseline for Rank, suggesting the effectiveness and potential of Glimpse.

## 3.3 MAIN RESULTS

**Accuracy and Efficiency.** We first compare Glimpse with existing black-box methods on their detection accuracy, speed, and cost. As Table 1 shows, Fast-Detect (GPT-3.5) surpasses both Likelihood (GPT-3.5) and DNA-GPT (GPT-3.5) with a significant margin in five source models (in AUROC). The basic method LogRank (GPT-3.5) also outperforms DNA-GPT in four-fifth source models, demonstrating the effectiveness and universality of Glimpse. Furthermore, Glimpse methods spend significantly less time and cost than DNA-GPT during the detection process. Specifically, DNA-GPT takes a total of 1911 seconds across the three datasets, while Glimpse takes just 462 seconds, making the detection process 4.1 times faster. Since DNA-GPT creates 10 completions per

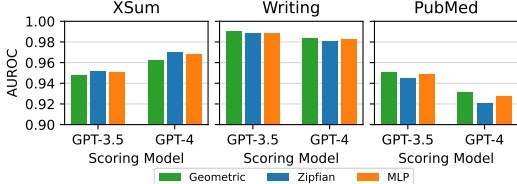 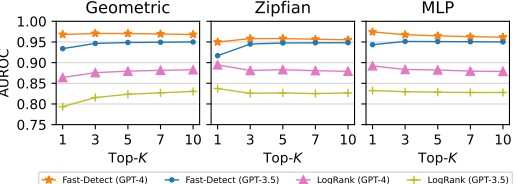

Figure 4: Ablation on *estimation algorithm*, where the AUROC is averaged across the five source models. Each dataset has its own preferred algorithm.

Figure 5: Ablation on *top-K*, where the AUROC is averaged across the datasets produced by GPT-4. Each line represents a combination of methods and scoring models.

passage, but Glimpse merely echos the probabilities of the input, DNA-GPT ends up costing about 10 times more than Glimpse based on current pricing (where the cost per output token is twice the input token)[5].

Here, we skip the comparison with DetectGPT in the black-box setting, because it requires 100x API calls and is affirmed to have lower accuracies compared to DNA-GPT (Yang et al., 2023a) and Fast-Detect (Bao et al., 2023), which are already our baselines.

**Latest LLMs are more accurate detectors.** We then compare our method with other existing methods. Powered by Glimpse, white-box methods such as Rank, LogRank, and Fast-Detect using GPT-3.5 significantly outperform their open-source versions using Neo-2.7, with a relative improvement of 41%, 21%, and 51%, respectively. Glimpse methods with smaller models also show competitive results. Fast-Detect (Babbage) outperforms Fast-Detect (Neo-2.7) on all five source models, even though Babbage (1.3B) is smaller than Neo-2.7 (2.7B). These results demonstrate that the latest LLMs with Glimpse are strong detectors.

**Larger LLMs can be universal detectors.** A recent study (Mireshghallah et al., 2023) reports that smaller LLMs are more efficient universal detectors, which can detect generations from various source models. However, our experimental results suggest that larger models can also be universal detectors, which perform better than smaller proprietary and open-source models in various source models. Specifically, GPT-3.5 (175B) achieves an average AUROC of about 0.95, outperforming smaller Babbage (1.3B) by about ↑ 43% and Neo-2.7 (2.7B) by about ↑ 51%. These findings suggest that larger LLMs, when equipped with the right technique, can also serve as universal detectors.

### 3.4 ABLATION STUDY

**Necessity of Glimpse.** Readers may wonder the necessity of these estimation algorithms, given that the top-$K$ probabilities provide the major information. To testify it, we consider a Naive approach to estimate the full distribution, where we assign zero probability to ranks larger than $K$. Using the Naive distribution, the average AUROC of Fast-Detect (GPT-3.5) is downgraded from 0.9630 (Geometric) to 0.9311 (Naive), indicating the necessity of a proper estimation algorithm.

**Ablation on Estimation Algorithm.** As the main results in Table 1 show, the Glimpses using Geometric, Zipfian, and MLP distributions do not show significant differences in their average accuracies. However, when we look into each dataset, we see different patterns as Figure 4 demonstrates. Specifically, the geometric distribution performs the best in Writing and PubMed, while Zipfian performs the best on XSum. MLP is more balanced, which performs the median on three datasets. These patterns remain with both GPT-3.5 and GPT-4 as the scoring model. These results suggest that different estimation algorithms may suit different situations, which is also supported by experiments on languages shown in Table 2.

**Ablation on Top-$K$.** Intuitively, the higher the value of $K$, the more precise the estimated distribution is likely to be. We conduct an ablation study with two representative methods with two scoring models as shown in Figure 5. The results, particularly with the geometric distribution, generally corroborate intuition. However, Zipfian and MLP show exceptions, where LogRank (GPT-4 and GPT-3.5) with Zipfian and Fast-Detect and LogRank (GPT-4) with MLP obtain their highest accuracies with the top-1 setting. The results indicate that even with limited top-1 probabilities, Glimpse can still work effectively when a proper estimation algorithm is used.

---

[5]https://azure.microsoft.com/en-us/pricing/details/cognitive-services/openai-service/

Figure 6: Robustness on *low false alarms*, where the red lines indicate false alarms of 1% and 10%. We draw the curves using Mix3 to simulate the real scenarios of using a single threshold to detect different domains. The dash lines denote the random classifier.

Table 2: Robustness over *languages*, where we use the same settings as the main experiments. Mix6 denotes the diverse mixture of the six languages.

| Method | Chinese (Web QA) | Russian (RuATD) | Urdu (News) | Indonesian (News) | Arabic (Wikipedia) | Bulgarian (News) | Mix6 |
|---|---|---|---|---|---|---|---|
| Fast-Detect (GPT-J/Neo-2.7B) | 0.9319 | 0.8158 | 0.9630 | 0.9876 | 0.9121 | 0.9422 | 0.8862 |
| Fast-Detect (Phi2-2.7B) | 0.8024 | 0.6710 | 0.9049 | 0.9313 | 0.9155 | 0.7500 | 0.8205 |
| Fast-Detect (Qwen2.5-7B) | 0.5345 | 0.7065 | 0.9970 | 0.9687 | 0.8404 | 0.8992 | 0.8063 |
| Fast-Detect (Llama3-8B) | 0.8729 | 0.7984 | 0.9962 | 0.9922 | 0.9282 | 0.9723 | 0.8713 |
| *Glimpse using Geometric* | | | | | | | |
| Fast-Detect (Babbage) | 0.9814 | 0.7673 | 0.9894 | 0.9857 | 0.9916 | 0.9628 | 0.8950 |
| Fast-Detect (Davinci) | **0.9938** | 0.8030 | 0.9996 | 0.9991 | 0.9982 | 0.9842 | 0.9369 |
| Fast-Detect (GPT-3.5) | 0.9913 | 0.8555 | **1.0000** | 0.9996 | **0.9999** | **0.9925** | **0.9774** |

**Ablation on Rank-List Size and Prompt.** The size of the rank list is a crucial hyper-parameter affecting Geometric and Zipfian distributions. Larger size typically results in higher accuracy, but it varies between estimation algorithms (see Appendix D.1). In addition, we examine how different prompts affect text detection accuracy in large models. The most sensitive model is GPT-4, in contrast to less sensitive GPT-3.5 and the least sensitive Babbage and Davinci (see Appendix D.2).

## 3.5 ANALYSIS AND DISCUSSION

**Robustness across Source Models and Domains.** In real scenarios, we tend to use a constant threshold to detect text from various sources and domains. We evaluate the stability of Glimpse and other baselines across the source models and datasets using thresholds found on "out-of-domain" datasets. The results show that Glimpse consistently provides the highest accuracy across the source models and domains, showcasing its stability. See Appendix E.1 for detailed setup and discussion.

**Robustness on Low False Alarms.** In real scenarios, an effective detector is expected to have a high recall (true positive rate) with a low false alarm (false positive rate), thereby allowing it to identify most machine-generated text without misclassifying human-written text. We evaluate the capacity of the methods by contrasting their ROC curves. As illustrated in Figure 6, for a false alarm in the range of $(0.01, 0.1)$, Fast-Detect (GPT-4) performs the best, except on ChatGPT generations, where Fast-Detect (Babbage) has the highest recalls. For a false alarm lower than 0.001, Fast-Detect (Babbage) performs consistently better than other methods, suggesting the advantage of it for low false alarm setting. Compared to baseline Likelihood (GPT-3.5) and Fast-Detect (Neo-2.7), Glimpse versions of Fast-Detect show a consistent advantage.

**Robustness over Languages.** We assessment Glimpse methods on M4 datasets with six languages. As Table 2 shows, Fast-Detect (GPT-3.5) consistently outperforms the baselines, as well as other proprietary LLMs. The system almost flawlessly identifies Urdu, Indonesian, and Arabic texts. However, the detection accuracy for Russian texts is much lower, indicating a potential under-training of the LLMs on this particular language. These findings imply that Glimpse with the latest LLMs can be effective and reliable detectors across languages.

**Robustness under Paraphrasing Attack.** We test the performance of Glimpse against paraphrasing attack using DIPPER, with two settings: high lexical diversity and high order diversity. Fast-Detect (Babbage) outperforms Fast-Detect (Neo-2.7) in both, but is more affected by diverse lexicons. An unusual behavior of DIPPER is observed in XSum, which causes an exceptionally low accuracy of the Likelihood (GPT-3.5) baseline. Glimpse surpasses Fast-Detect (Neo-2.7) and trained detectors,

proving its effectiveness. Larger LLMs are more vulnerable to increased diversity, which also reduce readability. See Appendix E.2 for a detailed comparison and discussion.

**Limitations.** Glimpse does not support all white-box methods, especially the rare methods that use inner embeddings instead of predictive distributions, such as PHD (Tulchinskii et al., 2024). Additionally, not all proprietary models provide Completion API, therefore, Glimpse cannot use them as the scoring model.

**Broader Impact.** The estimation methods can potentially be applied to other scenarios. For example, ChatCompletion API also provides top-$K$ probabilities. Using Glimpse, we can estimate the predictive distribution, which could be used to calculate some statistical metrics on the generated content. Such metrics could potentially be used to indicate, for instance, the level of hallucination in the content. Despite the effectiveness and potential of Glimpse, unsupervised use can lead to potential unfairness towards certain groups. For instance, detectors might display bias against non-native writers (Liang et al., 2023).

## 4 RELATED WORK

In terms of using a proprietary model for scoring, our method is in line with existing black-box methods. However, it inherits existing white-box approaches.

**Black-Box Methods for Zero-Shot Detection.** Zero-shot detection in the black-box setting is challenging due to restricted access of the model. DetectGPT (Mitchell et al., 2023) and its improvement NPR (Su et al., 2023) requires multiple evaluations of text sequences, while DNA-GPT (Yang et al., 2023a) requires multiple generations of text sequences, resulting in low speed and high cost. Another line of approaches treats detection as a question-answering task but does not reliably discern text generated by ChatGPT and GPT-4 (Bhattacharjee & Liu, 2024). These methods generally presume the knowing of the source model and use it to ascertain if a text was produced by it, which limits their usage to texts from uncertain origin. Unlike these approaches, our method minimizes the cost and dependence on known sources.

**White-Box Methods for Zero-Shot Detection.** Zero-shot methods in the white-box setting analyze a variety of metrics from model predictive distributions or output embeddings, including Entropy and Perplexity (Lavergne et al., 2008), Likelihood (Hashimoto et al., 2019; Solaiman et al., 2019), Rank and Log-Rank (Gehrmann et al., 2019), LRR (Su et al., 2023), Fast-DetectGPT (Bao et al., 2023), PHD (Tulchinskii et al., 2024), FourierGPT (Xu et al., 2024), and Binoculars (Hans et al., 2024). These methods, except that PHD requires output embeddings, can all be applied to proprietary models using Glimpse. However, in this study, we focus on very basic Entropy, Rank, and Log-Rank, together with recent Fast-DetectGPT, leaving the rest for future exploration.

**Other Detection Methods.** Trained classifiers are the majority of black-box methods, which rely human-authored texts along with LLM-generated content for training (Bakhtin et al., 2019; Uchendu et al., 2020; Solaiman et al., 2019; Ippolito et al., 2020; Fagni et al., 2021; Solaiman et al., 2019; Fagni et al., 2021; Yan et al., 2023; Li et al., 2023; Zeng et al., 2024b) (Verma et al., 2024; Kushnareva et al., 2024). However, they can become overly specialized in their training conditions, such as specific domains, languages, or source models. White-box approaches such as watermarking provide a different strategy, altering the LLM decoding process to include unique text signatures (Kirchenbauer et al., 2023; Kuditipudi et al., 2023; Christ et al., 2024; Zhao et al., 2023b;a). However, they require proactive text generation injection, which is not allowed by proprietary models. In this paper, we investigate techniques that do not need training or proactive injection.

## 5 CONCLUSION

We proposed Glimpse to estimate full distributions from partial observations, enabling white-box text detection methods to use proprietary models. Experiments show that Glimpse with the latest LLMs performs significantly better than its counterparts in terms of accuracy, efficiency, and robustness, highlighting the benefits of using the latest LLMs as detectors. Additional results in various white-box methods underscore the effectiveness of Glimpse, which may lead to a new direction of zero-shot detection in the black-box setting. To our knowledge, we are the first to enable white-box methods in proprietary LLMs.

ACKNOWLEDGEMENT

We would like to thank the anonymous reviewers for their valuable feedback. We thank Minjun Zhu for valuable discussion. This work is funded by the National Natural Science Foundation of China Key Program (Grant No. 62336006) and the Pioneer and "Leading Goose" R&D Program of Zhejiang (Grant No. 2022SDXHDX0003). Yanbin Zhao is supported by the National Natural Science Foundation of China (Grant No. 12201158).

ETHICAL STATEMENT

High efficient and accurate machine-generated text detector can potentially contribute to trustworthy AI techniques, which can mitigate the risks posed by the uncontrolled use of LLMs. Such technology can potentially benefit text readers, policy makers, and media platforms in a wide way.

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

# A GLIMPSE: A PROBABILITY DISTRIBUTION ESTIMATION APPROACH

As indicated in Figure 7a, the probability distribution across ranks generally follows a decaying pattern, where the larger models tend to have a higher top-1 probability and a bigger decay factor demonstrating a sharper distribution. We approximate the pattern using parameterized distributions, allocating the remaining probability mass (as '*' indicates) to ranks larger than $K$. We discuss three specific estimation algorithms with decaying patterns like Figure 7b.

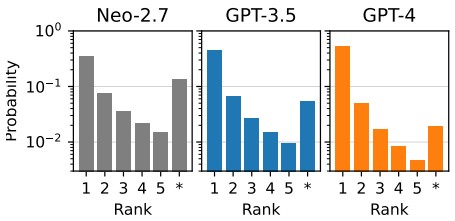 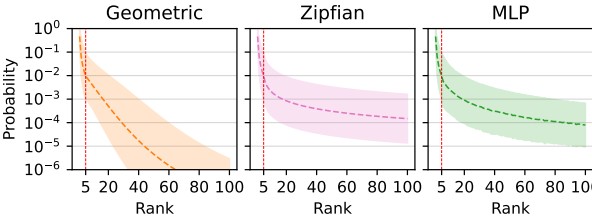

(a) *Top-5 probabilities*, where '*' indicates the remaining probability mass.

(b) *Full distribution* completing the top-5 probabilities from GPT-3.5, estimated using the three algorithms.

Figure 7: *Decaying patterns* of averaged probability distribution over ranks, where the probabilities are shown in log scale and evaluated on XSum.

## A.1 ESTIMATION USING GEOMETRIC DISTRIBUTION

As the simplest decaying pattern, we consider exponential decay with a fixed decay factor, resulting in a geometric distribution. Considering only the top-1 probability $p_1$, the whole distribution could be estimated from $p_1$ using

$$p(k) = p_1 \cdot (1 - p_1)^{k-1}, \text{ for } k \in [1..\infty], \tag{10}$$

where the probability decays with a factor of $\lambda = (1 - p_1)$. However, this standard geometric distribution only considers the probability of top-1 and is defined over infinite $k$, which is not suitable for a finite vocabulary.

Consequently, we extend the distribution to multiple top probabilities and a limited range of $k$. We express the probabilities for ranks larger than $K$ in near-Geometric distribution that

$$\begin{cases} p(k) = p_k, & \text{for } k \in [1..K] \\ p(k) = p_K \cdot \lambda^{k-K}, & \text{for } k \in [K+1..M] \\ \sum_{k=1}^{M} p(k) = 1, \end{cases} \tag{11}$$

where $\lambda$ is a decay factor in $(0, 1)$, and $M$ is the size of the rank list.

Using the total probability constraint, we calculate the remaining probability mass for allocating

$$\sum_{k=K+1}^{M} p(k) = 1 - \sum_{k=1}^{K} p_k = p_{\text{rest}}, \tag{12}$$

Expanding $p(k)$ in the left expression, we obtain

$$p_K \cdot \sum_{k=1}^{M-K} \lambda^k = p_{\text{rest}}, \tag{13}$$

and rewritten as

$$\sum_{k=1}^{M-K} \lambda^k = \frac{(\lambda - \lambda^{M-K+1})}{1 - \lambda} = \frac{p_{\text{rest}}}{p_K}. \tag{14}$$

Assuming $\lambda^{M-K+1}$ is close to zero, we reach an approximate solution

$$\lambda \approx \frac{p_{\text{rest}}}{p_K + p_{\text{rest}}}. \tag{15}$$

In practice, we first calculate the approximate solution, naming $\lambda_0$, and then check if $\lambda_0^{M-K+1}$ was close to zero. If it is not, we follow an iterative process to adjust $\lambda$ until it converges

$$\lambda_{t+1} = 1 - \frac{(\lambda_t - \lambda_t^{M-K+1}) \cdot p_K}{p_{\text{rest}}}. \tag{16}$$

## A.2 ESTIMATION USING ZIPFIAN DISTRIBUTION

Frequencies of words in natural languages usually adhere to Zipf's law (Zipf, 1946; 2013), where the word frequency and word rank follow a Zipfian distribution

$$p(k) \propto \frac{1}{(k+\beta)^\alpha}, \text{ for } k \in [1..\infty]. \tag{17}$$

The parameters $\alpha$ and $\beta$ are fitted to a specific corpus, with typical values of $\alpha = 1$ and $\beta = 2.7$ for English (Piantadosi, 2014).

Assuming that the word frequencies in a given context also comply with this law, we consider it as an alternative distribution for our estimation. Given the top-$K$ probabilities $p_k$, we compute the probabilities of tokens with a ranking greater than $K$ in a Zipfian distribution

$$\begin{cases} p(k) = p_k, & \text{for } k \in [1..K] \\ p(k) = p_K \cdot (\frac{\beta}{k-K+\beta})^\alpha, & \text{for } k \in [K+1..M] \\ \sum_{k=1}^M p(k) = 1, \end{cases} \tag{18}$$

where $\alpha$ and $\beta$ are two positive parameters. $M$ is the size of the rank list.

Using the total probability mass as a constraint, we determine the probability mass for allocating.

$$\sum_{k=K+1}^M p(k) = 1 - \sum_{k=1}^K p_k = p_{\text{rest}}, \tag{19}$$

After expanding $p(k)$, we get

$$p_K \cdot \sum_{k=K+1}^M (\frac{\beta}{k-K+\beta})^\alpha = p_{\text{rest}}, \tag{20}$$

rewritten as

$$\sum_{k=K+1}^M (\frac{\beta}{k-K+\beta})^\alpha = \sum_{k=1}^{M-K} (\frac{\beta}{k+\beta})^\alpha = \frac{p_{\text{rest}}}{p_K}. \tag{21}$$

The equation has two unknown parameters, thereby having multiple possible solutions. Thus, we solve the parameters by minimizing a loss function

$$\text{Loss}(\alpha, \beta) = \left( \sum_{k=1}^{M-K} (\frac{\beta}{k+\beta})^\alpha - \frac{p_{\text{rest}}}{p_K} \right)^2 + 1.0 \cdot (\alpha - 1)^2 + 0.001 \cdot (\beta - 2.7)^2. \tag{22}$$

As an additional constraint, we expect the parameters not vary from their typical values too much. We empirically determine a coefficient of $1.0$ for $\alpha$ and a coefficient of $0.001$ for $\beta$.

To accelerate the optimization process, we construct a table $T[\alpha, \beta]$, which stores the pre-calculated summation values for each pair of $\alpha$ and $\beta$ as

$$T[\alpha, \beta] = \sum_{k=1}^{M-K} (\frac{\beta}{k+\beta})^\alpha. \tag{23}$$

We enumerate $\alpha \in (0, 10)$ with a step of $0.1$ and $\beta \in (0, 20)$ with a step of $0.2$, resulting in a table with 10000 values. The ranges are empirically decided, which balance the coverage of the possible choices and the size of the table. During inference, we can efficiently compute the loss table $Loss(\alpha, \beta)$ from $T[\alpha, \beta]$ given $p_{\text{rest}}/p_K$. We then search the loss table to identify the best $\alpha$ and $\beta$ that lead to the smallest loss.

## A.3 ESTIMATION USING A MLP MODEL

The Geometric and Zipfian algorithms both work on assumptions about the distributions. An alternative approach that does not rely on these assumptions involves modeling the distribution within a neural network. We consider the simple multilayer perception model (MLP) with a single hidden layer, which accepts the top-$K$ probabilities and predicts the probabilities for the rest of the ranks. The distribution is expressed as

$$\begin{cases} p(k) = p_k, & \text{for } k \in [1..K] \\ p(k) = p_{\text{rest}} \cdot p_{\text{MLP}_\theta}(k - K), & \text{for } k \in [K + 1..M] \\ \sum_{k=1}^{M} p(k) = 1, \end{cases} \tag{24}$$

where $p_{rest} = 1 - \sum_{k=1}^{K} p_k$ and $p_{\text{MLP}_\theta}$ represents the MLP predictive distribution. The MLP is defined as

$$p_{\text{MLP}_\theta} = \text{SOFTMAX}(\text{MLP}_\theta(x)), \tag{25}$$

where $\theta$ denotes the model parameters. The model inputs a vector with a size of $K$ and outputs a distribution with a size of $M - K$. The input vector $x$ is calculated from the top probabilities using

$$x_k = \log p_k, \text{ for } k \in [1..K]. \tag{26}$$

**Training.** We train the MLP model on probability distributions from an open-source LLM (e.g., GPT-Neo-2.7B), using the cross-entropy loss

$$\text{Loss} = -\sum_{k=1}^{M} p_k \log p(k) = -\sum_{k=K+1}^{M} p_k \log p_{\text{MLP}_\theta}(k - K) + C, \tag{27}$$

where $p_k$ for $k \in [1..M]$ is the target distribution and $p(k)$ is the model distribution. $C$ is the constant part.

**Inference.** We predict the distributions using the top-$K$ probabilities from the proprietary LLMs, obtaining estimated distributions on all token positions. We do not enforce the monotonic decrease constraint during inference, but it generally follows the constraint because the training target is monotonic decrease distribution.

## B EXPERIMENTAL SETTINGS

### B.1 EVALUATION METRICS

**AUROC.** We measure the detection accuracy mainly in the area under the receiver operating characteristic (AUROC), which gives an overview of the detectors across all possible thresholds. AUROC values can range from 0.0 to 1.0, and this value mathematically signifies the probability that a randomly chosen machine-generated text has a higher predicted probability of being machine-generated compared to a randomly selected human-written text. An AUROC of 0.5 is indicative of a random classifier, while an AUROC of 1.0 suggests a flawless classifier.

**Accuracy (ACC).** As a complement, we report the ACC for some of the experiments. ACC denotes the ratio of the number of correct predictions to the total number of input samples, which works well only if there are equal number of positive and negative samples.

**True Positive Rate (TPR)** and **False Positive Rate (FPR).** In depth, we compare the methods in TPR versus FPR on various thresholds. A high TPR indicates that the algorithm is effective in identifying positive cases, while a high FPR indicates that the algorithm often misclassifies negative cases as positive.

### B.2 AZUREOPENAI SETTINGS

We use Completion API [6] [7] of these models via AzureOpenAI platform [8], with the following deployment settings. We echo the model to return the top probabilities of the provided texts, without producing any new tokens.

Different models are supported by different regions of Azure platform. We deploy babbage-002 and davinci-002 on the region of North Central US, gpt-35-turbo-0301 on the region of East US, and gpt-35-turbo-1106 and gpt-4-1106 on the region of West US. In addition, we also tried babbage-002 from the OpenAI API endpoint, obtaining the same results as the one on the AzureOpenAI endpoint.

## C  MAIN RESULTS

Table 3: Main results on *GPT-4 and Gemini-1.5* generations, with the best AUROC marked in **bold**.

| Method | GPT-4 | | | | Gemini-1.5 Pro | | | |
|---|---|---|---|---|---|---|---|---|
| | XSum | Writing | PubMed | Mix3 | XSum | Writing | PubMed | Mix3 |
| GPTZero | **0.9815** | 0.8838 | 0.8193 | 0.9009 | - | - | - | - |
| **Zero-Shot Detectors Using Open-Source LLMs** | | | | | | | | |
| Likelihood (Neo-2.7) | 0.7980 | 0.8553 | 0.8104 | 0.7690 | 0.8013 | 0.8364 | 0.7064 | 0.7416 |
| Entropy (Neo-2.7) | 0.4360 | 0.3702 | 0.3295 | 0.4114 | 0.4170 | 0.2945 | 0.3838 | 0.3959 |
| Rank (Neo-2.7) | 0.6644 | 0.7146 | 0.5965 | 0.6448 | 0.6711 | 0.6719 | 0.5688 | 0.6260 |
| LogRank (Neo-2.7) | 0.7975 | 0.8286 | 0.8003 | 0.7626 | 0.8022 | 0.8102 | 0.7006 | 0.7353 |
| DNA-GPT (Neo-2.7) | 0.7347 | 0.8032 | 0.7565 | 0.6430 | 0.7996 | 0.8133 | 0.6376 | 0.6438 |
| DetectGPT (T5-11B/Neo-2.7) | 0.5660 | 0.6217 | 0.6805 | 0.6136 | 0.7838 | 0.8256 | 0.6222 | 0.7406 |
| Fast-Detect (GPT-J/Neo-2.7) | 0.9067 | 0.9612 | 0.8503 | 0.8999 | 0.8571 | 0.8650 | 0.7075 | 0.8072 |
| Fast-Detect (Phi2-2.7B) | 0.4636 | 0.6463 | 0.6083 | 0.5742 | 0.6454 | 0.6324 | 0.5976 | 0.6164 |
| Fast-Detect (Qwen2.5-7B) | 0.6476 | 0.8202 | 0.6391 | 0.6862 | 0.7082 | 0.7723 | 0.6296 | 0.6839 |
| Fast-Detect (Llama3-8B) | 0.6615 | 0.8491 | 0.7556 | 0.7269 | 0.7786 | 0.9085 | 0.7065 | 0.7552 |
| **Zero-Shot Detectors Using Proprietary LLMs** | | | | | | | | |
| Likelihood (GPT-3.5) | 0.6468 | 0.9570 | 0.9152 | 0.8029 | 0.7130 | 0.9644 | 0.8516 | 0.8043 |
| DNA-GPT (GPT-3.5) | 0.7952 | 0.8302 | 0.9092 | 0.7748 | 0.8036 | 0.6829 | 0.7738 | 0.7107 |
| *Glimpse using Geometric* | | | | | | | | |
| Entropy (GPT-3.5) | 0.6353 | 0.2694 | 0.2376 | 0.4074 | 0.6237 | 0.2276 | 0.3361 | 0.4144 |
| Rank (GPT-3.5) | 0.6245 | 0.8719 | 0.8283 | 0.7395 | 0.6480 | 0.8793 | 0.7768 | 0.7406 |
| LogRank (GPT-3.5) | 0.6319 | 0.9323 | 0.9060 | 0.7870 | 0.7102 | 0.9421 | 0.8329 | 0.7872 |
| Fast-Detect (Babbage) | 0.9033 | 0.9264 | 0.9195 | 0.8974 | 0.8797 | 0.8316 | 0.7887 | 0.8083 |
| Fast-Detect (Davinci) | 0.9141 | 0.9798 | 0.8864 | 0.9131 | 0.9062 | 0.9516 | 0.7612 | 0.8601 |
| Fast-Detect (GPT-3.5) | 0.9035 | **0.9957** | 0.9467 | 0.9411 | 0.9221 | **0.9840** | **0.9112** | **0.9244** |
| Fast-Detect (GPT-4) | 0.9673 | 0.9901 | **0.9534** | 0.9647 | 0.9188 | 0.9506 | 0.8477 | 0.8947 |
| *Glimpse using Zipfian* | | | | | | | | |
| Fast-Detect (GPT-3.5) | 0.9123 | 0.9931 | 0.9429 | 0.9319 | 0.9289 | 0.9809 | 0.9070 | 0.9161 |
| Fast-Detect (GPT-4) | 0.9797 | 0.9884 | 0.9436 | **0.9719** | **0.9303** | 0.9447 | 0.8336 | 0.8991 |
| *Glimpse using MLP* | | | | | | | | |
| Fast-Detect (GPT-3.5) | 0.9076 | 0.9930 | 0.9464 | 0.9342 | 0.9257 | 0.9810 | 0.9103 | 0.9184 |
| Fast-Detect (GPT-4) | 0.9759 | 0.9893 | 0.9496 | 0.9705 | 0.9272 | 0.9495 | 0.8453 | 0.9001 |

---

[6] https://platform.openai.com/docs/guides/text-generation/completions-api

[7] Completion API with gpt-35-turbo-1106 and gpt-4-1106 require an AzureOpenAI API version of '2024-02-15-preview' or later, while others require '2023-09-15-preview' or later.

[8] https://azure.microsoft.com/en-us/products/ai-services/openai-service

Table 4: Main results on *Claude-3* generations, with the best AUROC marked in **bold**.

| Method | Claude-3-Sonnet | | | | Claude-3-Opus | | | |
|---|---|---|---|---|---|---|---|---|
| | XSum | Writing | PubMed | Mix3 | XSum | Writing | PubMed | Mix3 |
| **Zero-Shot Detectors Using Open-Source LLMs** | | | | | | | | |
| Likelihood (Neo-2.7) | 0.8862 | 0.9484 | 0.8360 | 0.8661 | 0.9322 | 0.9734 | 0.8603 | 0.9030 |
| Entropy (Neo-2.7) | 0.4146 | 0.2156 | 0.2989 | 0.3466 | 0.3871 | 0.1792 | 0.2910 | 0.3265 |
| Rank (Neo-2.7) | 0.7019 | 0.7812 | 0.6017 | 0.6888 | 0.7333 | 0.7950 | 0.6080 | 0.7056 |
| LogRank (Neo-2.7) | 0.8867 | 0.9401 | 0.8296 | 0.8654 | 0.9357 | 0.9679 | 0.8508 | 0.9042 |
| DNA-GPT (Neo-2.7) | 0.8558 | 0.9415 | 0.7647 | 0.7080 | 0.9424 | 0.9653 | 0.7806 | 0.7326 |
| DetectGPT (T5-11B/Neo-2.7) | 0.8150 | 0.8675 | 0.7347 | 0.7967 | 0.7718 | 0.8335 | 0.7752 | 0.7776 |
| Fast-Detect (GPT-J/Neo-2.7) | 0.9514 | 0.9763 | 0.8634 | 0.9260 | 0.9779 | 0.9832 | 0.8947 | 0.9468 |
| Fast-Detect (Phi2-2.7B) | 0.7536 | 0.6773 | 0.7144 | 0.6957 | 0.8080 | 0.7545 | 0.7322 | 0.7450 |
| Fast-Detect (Qwen2.5-7B) | 0.8595 | 0.8600 | 0.7346 | 0.7813 | 0.9097 | 0.8967 | 0.7572 | 0.8119 |
| Fast-Detect (Llama3-8B) | 0.9243 | 0.9198 | 0.7936 | 0.8212 | 0.9640 | 0.9377 | 0.8251 | 0.8510 |
| **Zero-Shot Detectors Using Proprietary LLMs** | | | | | | | | |
| Likelihood (GPT-3.5) | 0.8364 | 0.9918 | 0.9299 | 0.9023 | 0.9191 | 0.9955 | 0.9467 | 0.9295 |
| DNA-GPT (GPT-3.5) | 0.7934 | 0.8587 | 0.8988 | 0.7871 | 0.9040 | 0.9362 | 0.8926 | 0.8383 |
| *Glimpse using Geometric* | | | | | | | | |
|     Entropy (GPT-3.5) | 0.4685 | 0.0565 | 0.1985 | 0.2582 | 0.4310 | 0.0381 | 0.1852 | 0.2339 |
|     Rank (GPT-3.5) | 0.7801 | 0.9724 | 0.8476 | 0.8473 | 0.8263 | 0.9809 | 0.8524 | 0.8645 |
|     LogRank (GPT-3.5) | 0.8502 | 0.9927 | 0.9265 | 0.9062 | 0.9302 | **0.9966** | 0.9414 | 0.9336 |
|     Fast-Detect (Babbage) | 0.9508 | 0.9705 | 0.9111 | 0.9438 | 0.9874 | 0.9865 | 0.9298 | 0.9698 |
|     Fast-Detect (Davinci) | **0.9659** | **0.9939** | 0.9084 | 0.9606 | 0.9940 | 0.9946 | 0.9262 | 0.9742 |
|     Fast-Detect (GPT-3.5) | 0.9433 | 0.9930 | **0.9552** | 0.9576 | 0.9899 | 0.9829 | **0.9686** | 0.9689 |
|     Fast-Detect (GPT-4) | 0.9523 | 0.9910 | 0.9424 | 0.9623 | 0.9930 | 0.9917 | 0.9580 | **0.9817** |
| *Glimpse using Zipfian* | | | | | | | | |
|     Fast-Detect (GPT-3.5) | 0.9455 | 0.9920 | 0.9510 | 0.9475 | 0.9914 | 0.9798 | 0.9627 | 0.9588 |
|     Fast-Detect (GPT-4) | 0.9581 | 0.9889 | 0.9308 | 0.9613 | **0.9958** | 0.9901 | 0.9496 | 0.9792 |
| *Glimpse using MLP* | | | | | | | | |
|     Fast-Detect (GPT-3.5) | 0.9457 | 0.9925 | 0.9548 | 0.9526 | 0.9911 | 0.9811 | 0.9664 | 0.9634 |
|     Fast-Detect (GPT-4) | 0.9574 | 0.9901 | 0.9382 | **0.9631** | 0.9953 | 0.9909 | 0.9543 | 0.9807 |

Table 5: *A comparison of ACC* in Glimpse and the major baselines on generations from *ChatGPT*, where we employ the optimal threshold either for each dataset or across all three datasets. The smaller average drop scales on Glimpse methods indicate that Glimpse offers a more consistent metric across datasets. In addition, we also assess optimal threshold across datasets and source models, which yields ACCs nearly identical (deviation less than 0.006 except Likelihood and DNA-GPT) to the optimal threshold across datasets.

| Method | Best Threshold per Dataset | | | | Best Threshold across Datasets | | | | Drop |
|---|---|---|---|---|---|---|---|---|---|
| | XSum | Writing | PubMed | Avg. | XSum | Writing | PubMed | Avg. | Avg. |
| Fast-Detect (GPT-J/Neo-2.7) | 0.9600 | 0.9633 | 0.8267 | 0.9167 | 0.9400 | 0.9367 | 0.7567 | 0.8778 | -0.0389 |
| Fast-Detect (Phi2-2.7B) | 0.7467 | 0.6967 | 0.7600 | 0.7344 | 0.6767 | 0.6967 | 0.7467 | 0.7067 | -0.0277 |
| Fast-Detect (Qwen2.5-7B) | 0.7367 | 0.7600 | 0.7267 | 0.7411 | 0.6433 | 0.7600 | 0.6867 | 0.6967 | -0.0444 |
| Fast-Detect (Llama3-8B) | 0.8000 | 0.7700 | 0.7267 | 0.7656 | 0.6900 | 0.7667 | 0.6567 | 0.7044 | -0.0612 |
| Likelihood (GPT-3.5) | 0.8767 | 0.9933 | 0.8933 | 0.9211 | 0.7533 | 0.9900 | 0.8633 | 0.8689 | -0.0522 |
| DNA-GPT (GPT-3.5) | 0.8750 | 0.8592 | 0.8833 | 0.8725 | 0.8108 | 0.5986 | 0.7467 | 0.7187 | -0.1538 |
| *Glimpse using Geometric* | | | | | | | | | |
|     Fast-Detect (Babbage) | 0.9600 | 0.9433 | 0.9033 | 0.9356 | **0.9600** | 0.9100 | 0.8833 | 0.9178 | -0.0178 |
|     Fast-Detect (Davinci) | **0.9667** | 0.9800 | 0.8867 | 0.9444 | 0.9333 | 0.9700 | 0.8633 | 0.9222 | -0.0222 |
|     Fast-Detect (GPT-3.5) | 0.9633 | **0.9967** | **0.9200** | **0.9600** | 0.9033 | **0.9967** | **0.9167** | **0.9389** | -0.0211 |
|     Fast-Detect (GPT-4) | 0.9367 | 0.9733 | 0.8933 | 0.9344 | 0.9367 | 0.9400 | 0.8833 | 0.9200 | **-0.0144** |

# D ABLATION STUDY

## D.1 ABLATION ON RANK-LIST SIZE

The size $M$ of the rank list is another important hyperparameter. We assess its effects on Geometric and Zipfian distributions (skipping MLP because it requires a heavy training process for each setting). As demonstrated in Figure 8, overall Glimpse with a larger size obtains a higher accuracy. Geometric distribution shows a monotonic increasing trends, while Zipfian shows decreasing trends for Fast-Detect but increasing trends for LogRank, demonstrating an inconsistent pattern. Roughly, experiments with MLP on the sizes of 100 and 1000 suggest that it has the similar pattern as Zipfian.

Table 6: The prompts that we test for the ablation, from the empty prompt0 to simple prompt1 until complex prompt3 and prompt4. The changes are marked in *italic*.

| Prompt | Content |
|--------|---------|
| prompt0 | (Empty) |
| prompt1 | *You serve as a valuable aide, capable of generating clear and persuasive pieces of writing given a certain context. Now, assume the role of an author and strive to finalize this article.* |
| prompt2 | You serve as a valuable aide, capable of generating clear and persuasive pieces of writing given a certain context. Now, assume the role of an author and strive to finalize this article. *I operate as an entity utilizing GPT as the foundational large language model. I function in the capacity of a writer, authoring articles on a daily basis. Presented below is an example of an article I have crafted.* |
| prompt3 | *System:* You serve as a valuable aide, capable of generating clear and persuasive pieces of writing given a certain context. Now, assume the role of an author and strive to finalize this article. *Assistant:* I operate as an entity utilizing GPT as the foundational large language model. I function in the capacity of a writer, authoring articles on a daily basis. Presented below is an example of an article I have crafted. |
| prompt4 | *Assistant:* You serve as a valuable aide, capable of generating clear and persuasive pieces of writing given a certain context. Now, assume the role of an author and strive to finalize this article. *User:* I operate as an entity utilizing GPT as the foundational large language model. I function in the capacity of a writer, authoring articles on a daily basis. Presented below is an example of an article I have crafted. |

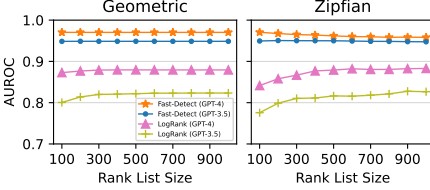

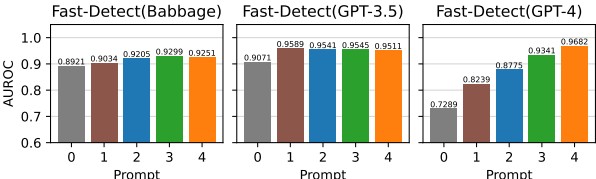

Figure 8: Ablation on *rank-list size*, where the AUROC is averaged across the three datasets produced by GPT-4.

Figure 9: Ablation on *prompt*, where the AUROC is averaged across the three datasets produced by GPT-4. GPT-4 is most sensitive to prompts.

## D.2 ABLATION ON PROMPT

Large models are sensitive to text context, necessitating a suitable prompt for optimal detection accuracy (Taguchi et al., 2024). We analyze the prompts featured in Table 6 to ascertain their effect. We draft the prompts manually, starting with prompt3. Then we replace the '*System*' and '*Assistant*' roles with '*Assistant*' and '*User*' to produce prompt4 and we remove the roles to produce prompt2. We simplify prompt2 by removing its second paragraph to produce prompt1 and removing all content to produce empty prompt0. In this study, we only experiment with several manually drafted prompts, leaving a systematic exploration of the prompts for the future.

As Figure 9 shows, GPT-4 is the most sensitive model among the four scoring models, with detection accuracy fluctuating between 0.7289 (prompt0) and 0.9682 (prompt4). In contrast, GPT-3.5 is less sensitive, with a detection accuracy increasing from 0.9071 (prompt0) to 0.9589 (prompt1), but maintaining stability for the rest. The base model Babbage and Davinci are less influenced by the prompt, and we do not show them in the figure.

Table 7: Robustness across *source models* measured in ACC, where we evaluate all source models using a threshold determined according to ChatGPT. All experiments are run on *Mix3*.

| Method | ChatGPT | GPT-4 | Claude-3 | | Gemini-1.5 | All |
| --- | --- | --- | --- | --- | --- | --- |
| | | | Sonnet | Opus | Pro | Avg. |
| Fast-Detect (GPT-J/Neo-2.7) | 0.8778 | 0.7933 | 0.8389 | 0.8822 | 0.6867 | 0.8158 |
| Fast-Detect (Phi2-2.7B) | 0.7067 | 0.5500 | 0.6378 | 0.6622 | 0.6011 | 0.6316 |
| Fast-Detect (Qwen2.5-7B) | 0.6967 | 0.6222 | 0.6811 | 0.7011 | 0.6256 | 0.6653 |
| Fast-Detect (Llama3-8B) | 0.7044 | 0.6556 | 0.7244 | 0.7467 | 0.6844 | 0.7031 |
| Likelihood (GPT-3.5) | 0.8689 | 0.6806 | 0.8326 | 0.8605 | 0.6703 | 0.7826 |
| DNA-GPT (GPT-3.5) | 0.7187 | 0.6176 | 0.6942 | 0.7540 | 0.6372 | 0.6843 |
| *Glimpse using Geometric* | | | | | | |
| Fast-Detect (Babbage) | 0.9178 | 0.7373 | 0.8461 | 0.9176 | 0.6562 | 0.8150 |
| Fast-Detect (Davinci) | 0.9222 | 0.8085 | 0.8973 | 0.9131 | 0.7363 | 0.8555 |
| Fast-Detect (GPT-3.5) | **0.9389** | 0.8650 | 0.8985 | 0.9197 | **0.8554** | **0.8955** |
| Fast-Detect (GPT-4) | 0.9200 | **0.9043** | **0.9041** | **0.9276** | 0.7973 | 0.8906 |

Table 8: Robustness across *domains* measured in ACC, where we cross-validate each dataset using a threshold determined according to other two datasets for each source model.

| Method | ChatGPT | | | | GPT-4 | | | |
| --- | --- | --- | --- | --- | --- | --- | --- | --- |
| | XSum | Writing | PubMed | Avg. | XSum | Writing | PubMed | Avg. |
| Fast-Detect (GPT-J/Neo-2.7) | 0.8633 | 0.9300 | 0.7000 | 0.8311 | 0.7867 | 0.8833 | 0.6600 | 0.7767 |
| Fast-Detect (Phi2-2.7B) | 0.6800 | 0.6967 | 0.7467 | 0.7078 | 0.4767 | 0.6267 | 0.5667 | 0.5567 |
| Fast-Detect (Qwen2.5-7B) | 0.6200 | 0.7300 | 0.6167 | 0.6556 | 0.5733 | 0.7433 | 0.5500 | 0.6222 |
| Fast-Detect (Llama3-8B) | 0.6567 | 0.7500 | 0.5767 | 0.6611 | 0.5967 | 0.6933 | 0.5800 | 0.6233 |
| Likelihood (GPT-3.5) | 0.6933 | 0.9833 | 0.8233 | 0.8333 | 0.5503 | 0.8020 | 0.8300 | 0.7274 |
| DNA-GPT (GPT-3.5) | 0.5000 | 0.5106 | 0.7067 | 0.5724 | 0.4966 | 0.5000 | 0.6933 | 0.5633 |
| *Glimpse using Geometric* | | | | | | | | |
| Fast-Detect (Babbage) | 0.9133 | 0.9100 | **0.8800** | 0.9011 | 0.8154 | 0.8033 | 0.8333 | 0.8174 |
| Fast-Detect (Davinci) | 0.8767 | 0.9633 | 0.8500 | 0.8967 | 0.8087 | 0.7800 | 0.8033 | 0.7974 |
| Fast-Detect (GPT-3.5) | 0.8833 | **0.9967** | 0.8267 | **0.9022** | 0.7819 | **0.9396** | **0.8667** | 0.8627 |
| Fast-Detect (GPT-4) | **0.9367** | 0.9367 | 0.8167 | 0.8967 | **0.9200** | 0.9195 | 0.8467 | **0.8954** |

# E ANALYSIS AND DISCUSSION

## E.1 ROBUSTNESS ACROSS SOURCE MODELS AND DOMAINS

In practice, we need to fix the decision threshold and detect text from various sources and domains. However, the distributions of the detection metric might be shifted between different sources or domains, resulting in high detection accuracy in one but low accuracy in another. In this section, we evaluate the robustness of Glimpse along with other strong baselines across different source models and domains.

Firstly, we examine the detection accuracy (in ACC) for each source model utilizing an optimal threshold identified in the ChatGPT Mix3 dataset. As illustrated in Table 7, Glimpse consistently provides the highest ACCs across all source models. While the accuracy of Fast-Detect (GPT-3.5) and Fast-Detect (GPT-4) fluctuates between source models, their overall ACCs are closely matched. Fast-Detect (GPT-3.5) delivers the highest ACC, which is approximately 8 points above the top baseline. This demonstrates the stability of Glimpse across numerous source models.

Subsequently, we assess the accuracy on each dataset using an optimal threshold established on the remaining two datasets for each source model. As exhibited in Table 8, we employ the source models of ChatGPT and GPT-4 as examples. Glimpse also delivers the highest ACCs on all datasets, further evidence of its robustness across various domains.

## E.2 ROBUSTNESS UNDER PARAPHRASING ATTACK

We assess the performance of Glimpse uder paraphrasing attack, utilizing DIPPER (Krishna et al., 2024) to rephrase the output generated by ChatGPT. Our testing encompasses two paraphrasing settings: high lexical diversity (60 L) and high-order diversity (60 O). As indicated in Table 9, Fast-Detect (Babbage) surpasses Fast-Detect (Neo-2.7) in both settings, but is more significant influenced by diverse lexicons than by diverse orderings.

Table 9: Robustness under *paraphrasing attack* with diverse lexicons and orders, where we report the TPR (%) at an FPR level of 1%.

| Method | ChatGPT + DIPPER (60 L) | | | | ChatGPT + DIPPER (60 O) | | | |
|---|---|---|---|---|---|---|---|---|
| | XSum | Writing | PubMed | Avg. | XSum | Writing | PubMed | Avg. |
| Fast-Detect (GPT-J/Neo-2.7) | 48.7 | **65.3** | 38.0 | 50.7 | 52.7 | 66.0 | 34.0 | 50.9 |
| Fast-Detect (Phi2-2.7B) | 6.7 | 15.3 | 8.0 | 10.0 | 0.7 | 0.7 | 5.3 | 1.8 |
| Fast-Detect (Qwen2.5-7B) | 16.0 | 35.3 | 8.0 | 19.8 | 5.3 | 10.7 | 6.0 | 7.3 |
| Fast-Detect (Llama3-8B) | 34.0 | 50.0 | 13.3 | 32.4 | 17.3 | 12.7 | 10.7 | 13.6 |
| Likelihood (GPT-3.5) | 0.7 | 11.3 | 9.3 | 6.9 | 0.7 | 71.3 | 18.7 | 30.2 |
| *Glimpse using Geometric* | | | | | | | | |
| Fast-Detect (Babbage) | **64.0** | 58.7 | **39.3** | **54.0** | **83.3** | 80.0 | **42.0** | **68.4** |
| Fast-Detect (Davinci) | 16.7 | 48.0 | 28.7 | 31.1 | 51.3 | 88.7 | 33.3 | 57.8 |
| Fast-Detect (GPT-3.5) | 0.7 | 62.0 | 12.0 | 24.9 | 18.0 | **95.3** | 22.7 | 45.3 |

However, we also note unusual behavior of DIPPER on XSum, where the diverse lexical paraphrasing results in a surprisingly low detection accuracy for Likelihood (GPT-3.5) baseline. This is caused by the atypical trend that the paraphraser replaces common words with rare expressions, thus reducing the readability of the content. This odd distribution could be the cause of the unusually low accuracy of Fast-Detect (GPT-3.5) on XSum (60 L). Glimpse outperforms the baselines with open source models, proving its efficacy in withstanding a paraphrasing attack.

Additionally, we find that larger LLMs are more susceptible to increased lexical and order diversity. Nonetheless, our review of the paraphrased articles indicates that this increased diversity also lowers the readability due to excessive use of unusual words and sequences, implying that there are some drawbacks of the attack.

