# OpenReview forum: "Glimpse: Enabling White-Box Methods to Use Proprietary Models for Zero-Shot LLM-Generated Text Detection"
_ICLR.cc/2025/Conference — ICLR 2025 Poster_

### Official Review · Reviewer_qGG2 · 2024-10-25

**Soundness:** 3
**Presentation:** 3
**Contribution:** 3
**Rating:** 8
**Confidence:** 3

**Summary:**

To enable existing white-box methods to proprietary models, the authors propose Probability Distribution Estimation (PDE), extending methods like Entropy, Rank, Log-Rank, and Fast-DetectGPT to latest closed-source LLMs. This work uses only the top-K probabilies returns by the API to estimate the necessary metrics (such as mean and variance in DetectGPT). They estimate the probs after rank K using parametrized distribution (which token each pk refers to is not important). Three algorithms are tested: Geometric, Zipfian, and MLP.

It is applied to ChatGPT (gpt-35-turbo) (OpenAI, 2022), GPT-4 (gpt-4) (OpenAI, 2023), Claude3 Sonnet (claude-3-sonnet-20240229) and Opus (claude-3-opus-20240229) (Anthropic, 2024), Gemini-1.5 Pro (gemini-1.5-pro). The results is quite promising, and suggests larger LLMs can be universal detectors.

**Strengths:**

I think the idea of estimating unknown probs using Geometric, Zipfian, and MLP is very interesting.

The idea can be applied to different white-box methods.

The performance are quite promising.

**Weaknesses:**

Comparing to this proprietary and estimation approach, why not use open-source large model such as LLaMA as the scoring model? This paper did not mention models like LLaMA at all. If authors can give me a convincing answer or add some comparison experiment, I'll consider raise score.

It still assumes some partial information from the API. But I do agree this setting is realistic and interesting.

Some parts of the writing can be improved (refer to my questions).

**Questions:**

Line302 : I did not understand what's the difference between source model and scoring models… My current understanding is that the authors want to argue that a good scoring model can detect machine-generated text from a wide range of LLMs, is my understanding correct? I think if you can add some text to explain that will be helpful.

Figure1: I think the notation p(x|x) is weird.

Line293: What do you mean by "generate corresponding text from source LLMs for each sample"? Do you mean generate text with the same prefix?

---

> ### Author Response · Authors · 2024-11-18
> **Reply 1 out of 1**
>
> We appreciate your detailed and thoughtful feedback, particularly your acknowledgement of the value of the task setting! Your concern W1 has been carefully examined and addressed in the revised version. We hope this resolves your issue. Please let us know if you have any further questions.
>
> ---
> > **W1:** Comparing to this proprietary and estimation approach, why not use open-source large model such as LLaMA as the scoring model? This paper did not mention models like LLaMA at all. If authors can give me a convincing answer or add some comparison experiment, I'll consider raise score.
>
> **Reply:** Please refer to the reply to Common Concern 2.
>
> ---
> > **Q1:** what's the difference between source model and scoring model.
>
> **Reply:** We have updated the paper for better clarity. The text is generated using the source model, and the scoring model is used to create a detection metric. In our tests, we utilize a consistent scoring model to identify text generated by different source models.
>
> ---
> > **Q2:** Figure1: I think the notation p(x|x) is weird.
>
> **Reply:** We have amended the paper to eliminate any confusion. The notation simply adheres to the format of the conditional probability function p(\tilde{x}|x) and considers p(x)= p(x|x) as a particular instance of the function.
>
> ---
> > **Q3:** Line293: What do you mean by "generate corresponding text from source LLMs for each sample"? Do you mean generate text with the same prefix?
>
> **Reply:** Indeed, you are correct. We have made revisions to the paper for better clarity.

---

> > ### Comment · Reviewer_qGG2 · 2024-11-19
> > **thank you, score updated.**
> >
> > The response has cleared my major concern, and I have updated my score to accept.
> > I think the concerns from other reviewers are also reasonable, but I personally find the approach proposed is very interesting.

---

> ### Author Response · Authors · 2024-11-20
> **Appreciate**
>
> We are profoundly grateful for your acknowledgment of our work, which will undoubtedly inspire our ongoing research!

---

### Official Review · Reviewer_NxVa · 2024-10-30

**Soundness:** 2
**Presentation:** 3
**Contribution:** 2
**Rating:** 6
**Confidence:** 3

**Summary:**

The paper proposes Probability Distribution Estimation (PDE), a method for enhancing the capabilities of white-box text detection methods, such as Fast-DetectGPT, in identifying machine-generated text from proprietary large language models (LLMs) like GPT-4 and Claude-3. PDE extends traditional white-box methods to proprietary models by estimating full probability distributions from limited API access. The authors demonstrate that using PDE with Fast-DetectGPT achieves improvements in detection accuracy over previous methods.

**Strengths:**

- A thorough explanation of the PDE approach, describing its implementation across different parameterization methods (e.g., Geometric, Zipfian).
- Extensive ablation studies and analyses (distribution choices, top-K probabilities, language variations). This adds depth and clarity to the method’s application across proprietary LLMs.
- Completing datasets with new models that authors will release

**Weaknesses:**

- The evaluations are conducted on limited datasets (150 pairs), potentially restricting the findings’ generalizability across more diverse or larger data sources. There are more samples for other models in the original M4 dataset.
- While PDE is an effective enhancement, it primarily focuses on density estimation and integration with existing detection methods rather than introducing a fundamentally new detection approach.
-  Although PDE improves upon Fast-Detect, the accuracy increase is relatively modest, raising questions about the practical advantage of this added complexity in real-world applications. In table 2 you report Fast-Detect (GPT-J/Neo-2.7) which has relatively good. It hints that it may be better to use bigger or newer open models (like Qwen2.5-1.5B or Llama3) than to use PDE to proprietary models.

**Questions:**

- Did you try to Fast-Detect with other open models rather than GPT-J/Neo-2.7? Could you please scores from newer/larger models for better comparison?
- The evaluation dataset is very small; did you try other datasets? I believe it would be better to perform testing on bigger datasets, for example the original M4.

Small remarks:
- line 101-102: "(↑ more than 25%)" might confuse readers. Better to add "relative to the remaining space"
- "*" in equations looks strange
- line 297: gpt-35-turbo -> gpt-3.5-turbo

---

> ### Author Response · Authors · 2024-11-18
> **Reply 1 out of 1**
>
> We appreciate your feedback. Your concern about W3 have been taken into consideration and addressed. However, we believe that your issues with W1 and W2 may be due to unfamiliarity with the recent studies about zero-shot machine-generated text detection. We hope our explanation will dispel your uncertainties. If you have additional questions, please don't hesitate to let us know.
>
> ---
> > **W1:** The evaluations are conducted on limited datasets (150 pairs), potentially restricting the findings’ generalizability across more diverse or larger data sources.
>
> **Reply:** Please refer to the reply to Common Concern 1.
>
> ---
> > **W2:** While PDE is an effective enhancement, it primarily focuses on density estimation and integration with existing detection methods rather than introducing a fundamentally new detection approach.
>
> **Reply:** PDE facilitates the application of white-box techniques on proprietary models, eliminating the previous restriction that only black-box methods could utilize proprietary models for detection. In terms of its impact, we believe PDE holds greater significance than a solitary detection method, as it allows for the integration of white-box methods with the most recent LLMs.
>
> ---
> > **W3:** it may be better to use bigger or newer open models (like Qwen2.5-1.5B or Llama3) than to use PDE to proprietary models.
>
> **Reply:** Please refer to the reply to Common Concern 2.
>
> ---
> > **Remarks:** Small remarks
>
> **Reply:** We have revised the paper for the descriptions and equations.

---

> > ### Author Response · Authors · 2024-11-21
> > **A kind enquiry**
> >
> > Dear Reviewer, we have diligently attended to your previously raised concerns. Would you mind providing us with any additional feedback you might have?
> >
> > Since our last response, we have incorporated additional experimental findings. These include a comparison of ACC in Table 6, an assessment of robustness across source models and domains in Appendix E.1, and modifications to TPR/FPR in Table 10.

---

> > > ### Comment · Reviewer_NxVa · 2024-11-25
> > >
> > > Thank you for your detailed response and for addressing the concerns raised in the initial feedback. I apologize for the delay in providing this update. It is clear that significant effort has been made to improve the paper, with a substantial number of new experiments conducted. I particularly appreciate the inclusion of evaluations with newer models, which add valuable context and robustness to the study.
> > >
> > > Based on the improvements and added contributions, I am raising my score.

---

> > > > ### Author Response · Authors · 2024-11-26
> > > > **Appreciate**
> > > >
> > > > We deeply appreciate your recognition of our work, which will certainly motivate us in our future research!

---

### Official Review · Reviewer_Gnc2 · 2024-10-31

**Soundness:** 2
**Presentation:** 2
**Contribution:** 2
**Rating:** 5
**Confidence:** 5

**Summary:**

The paper considers the topic of detecting the texts, generated by closed (proprietary) LLMs. Authors suggest several ways to approximate the distribution of token probabilities of those closed LLMs accessible via API (such as GPT-4), knowing only the probabilities of top-K or even top-1 tokens. Using these distributions, authors managed to emulate several methods of LLM-generated text detection (i. e. Fast-DetectGPT, Entropy, etc) as if they had access to these LLMs. Using these methods upon approximated distributions, they overperformed analogous methods based on distributions of GPT-Neo-2.7B.

**Strengths:**

- The topic of LLM-generated text detection is important.
- The idea of the paper is worth considering.
- Authors check how their detectors behave when they fix a low False Positive Rate, that is a good practice.
- Authors check their detectors against paraphrasing attacks and consider the influence of various prompts.

**Weaknesses:**

The major drawbacks of the paper lie in its experimental setup.
- The amount of machine-generated texts in the datasets in Table 2 (which is supposed to contain the main results) is extremely small. E.g., as far as I understood from part 3.1 SETTINGS, for each sub-domain "XSum", "Writing", and "PubMed" in Table 2, only 150 machine-generated texts vs 150 human-written texts are used. It's hard to tell anything for sure from such a small amount of examples, and the results may be very noisy. The authors need to increase their datasets to get reliable results.
- The choice of baseline open LLM (i.e. GPT-Neo-2.7B) for applying methods like Fast-Detect and Likelihood seems unreasonable because GPT-Neo-2.7B is weak compared to the newer open LLMs of comparable size. I would recommend adding at least the Phi-2 model (it also has 2.7B parameters) as a base for Fast-Detect, Likelihood, and other similar methods because its perplexity scores seem to be good in LLM-generated text detection (AI-generated text boundary detection with RoFT, Kushnareva et. al, 2024). Besides, many bigger models look like fair candidates for a good baseline (e.g. LLaMA 3.2 and many others).
- AUROC scores in the Tables seem to be very high both for the best baseline and for the best method suggested by the authors. It means that the task statement is too easy. It would be good to add more complicated data, where all the methods will struggle, and the difference between methods will become more clear.

Minor remarks about the presentation:
- Overusing the appendices: appendices D.1 and E look more suitable for the main text;
- Typos: log p(x|x) in Figure 1 and Equation (2);
- Figure 1 contains an extremely bright red color, it is hard to look at it. Please, change it to a softer color.

**UPDATE**: The first two of my concerns were addressed, so I updated my main score from 3 to 5. I adjusted scores about "soundness" and "presentation" accordingly.

**Questions:**

- Line 138-140, Equation 2: why do you call this $d(x, p_\theta)$ value "curvature"? It doesn't seem to have a meaning of a curvature. Maybe "skewness" would be better word?
- Only AUROC is reported as a metric for comparison in the Tables. It would be interesting to see, what are the accuracy scores for all methods (for some fixed threshold)?
- Are there any differences between "Likelihood" method and perplexity scores?

---

> ### Author Response · Authors · 2024-11-18
> **Reply 1 out of 1**
>
> Thank you for your valuable input. We have addressed your concern regarding W2. However, we suspect that your concerns with W1 and W3 might be due to a lack of familiarity with the task of identifying machine-generated text using zero-shot techniques. We hope our response will clarify your doubts. Please let us know if you have any further queries.
>
> ---
> > **W1:** The amount of machine-generated texts in the datasets in Table 2 (which is supposed to contain the main results) is extremely small.
>
> **Reply:** Please refer to the reply to Common Concern 1.
>
> ---
> > **W2:** The choice of baseline open LLM (i.e. GPT-Neo-2.7B) for applying methods like Fast-Detect and Likelihood seems unreasonable because GPT-Neo-2.7B is weak compared to the newer open LLMs of comparable size.
>
> **Reply:** Please refer to the replay to Common Concern 2.
>
> ---
> > **W3:** AUROC scores in the Tables seem to be very high both for the best baseline and for the best method suggested by the authors.
>
> **Reply:** It is commonplace for zero-shot detection techniques to achieve AUROC scores exceeding 0.90. This can be seen in recent research like DetectGPT by Mitchell et al. (ICML 2023), Fast-DetectGPT by Bao et al. (ICLR 2024), DNA-GPT by Yang et al. (ICLR 2024), and DLAD by Zeng et al. (NeurIPS 2024).
>
> ---
> > **Q1:** Line 138-140, Equation 2: why do you call this d(x,pθ) value "curvature"? It doesn't seem to have a meaning of a curvature. Maybe "skewness" would be better word?
>
> **Reply:** The idea of "probability curvature" was introduced by DetectGPT (Mitchell et al., ICML 2023), which is a measure of the difference between the log-likelihoods of the revised and original text. Machine-generated text typically has a positive curvature because its rephrased versions usually have lower log-likelihoods. However, human-written text often has a curvature close to zero because its rephrased versions can have either higher or lower log-likelihoods. Fast-DetectGPT adopts this idea but substitutes the "probability function" with the "conditional probability function". We mention Fast-DetectGPT here to demonstrate how PDE can be applied to it.
>
> ---
> > **Q2:** Only AUROC is reported as a metric for comparison in the Tables. It would be interesting to see, what are the accuracy scores for all methods (for some fixed threshold)?
>
> **Reply:** We also present the True Positive Rate against the False Positive Rate in Figure 6, enabling a clear comparison of the effectiveness of various methods at different thresholds. Furthermore, we believe that ACC is not a suitable metric for detecting machine-generated text as it depends on the proportion of positive to negative samples. Depending on the situation, the quantity of positive samples may be significantly greater or lesser than the negative samples, leading to inflated ACC scores even when using a detector that guesses randomly.
>
> ---
> > **Q3:** Are there any differences between "Likelihood" method and perplexity scores?
>
> **Reply:** Essentially, they are equivalent for zero-shot detection tasks. The likelihood represents the average log likelihood, calculated as "1/N ∑ log p(x_j | x_<j)", while the perplexity score is simply the exponential of the negative average log likelihood, expressed as "exp(-average log likelihood)". We will explain this further.
>
> ---
> > **Remarks:** Minor remarks about the presentation
>
> **Reply:** We will revise the paper to improve the presentation. Specifically, we will replace log p(x|x) with its equal expression log p(x), where log p(x|x) is just an expression in form of the conditional probability function. We will replace the color with a softer red. For appendix D.1 and E, we will consider putting them into the main text if space allows.

---

> ### Comment · Reviewer_Gnc2 · 2024-11-19
> **Response**
>
> Thank you for your answers, paper revision and new experiments. It was interesting and surprising that Neo-2.7B worked better as a backbone for your method than newer models.
>
> However, I find important to continue the discussion of the following point:
>
> > Q2: Only AUROC is reported as a metric for comparison in the Tables. It would be interesting to see, what are the accuracy scores for all methods (for some fixed threshold)?
>
> > Reply: We also present the True Positive Rate against the False Positive Rate in Figure 6, enabling a clear comparison of the effectiveness of various methods at different thresholds. Furthermore, we believe that ACC is not a suitable metric for detecting machine-generated text as it depends on the proportion of positive to negative samples. Depending on the situation, the quantity of positive samples may be significantly greater or lesser than the negative samples, leading to inflated ACC scores even when using a detector that guesses randomly.
>
> I think I need to elaborate my thoughts about AUROC and accuracy in more details. When the metrics are very close to each other, it becomes difficult to draw clear conclusions about the relative performance of the methods. And, as I already told in my review, there are several methods with high AUROC in your work.
>
> In the same time, accuracy is more discrete and more illustrative metric than AUROC. E.g. if you have 300 examples in your dataset, and the accuracy for a method A is bigger than the accuracy for a method B by 1%, then method A correctly classified three more examples compared to Method B (in general, the set of correctly identified examples can be different for two methods though). I asked to see the accuracy (for some fixed threshold) because it would be interesting to see exactly how many more examples your method classified correctly in each case.
>
> I completely agree that accuracy is not suitable metric when one is using unbalanced datasets. However, your datasets appear to be perfectly balanced (with an equal number of natural and generated samples), so in this case, accuracy can serve as a meaningful complement to the AUROC metric.
>
> ---
>
> I’m sorry that I don’t have the opportunity to address the other points right now, but I will respond to them later as well.

---

> ### Author Response · Authors · 2024-11-20
> **Reply to further questions regarding Q2.**
>
> We deeply appreciate your thorough explanation regarding the issue. We have updated the paper to incorporate a comparison of ACC in Table 6.
>
> Despite ACC not being commonly employed by recent zero-shot detection studies, such as DetectGPT (Mitchell et al., ICML 2023), Fast-DetectGPT (Bao et al., ICLR 2024), DNA-GPT (Yang et al., ICLR 2024), FourierGPT (Yang et al., EMNLP 2024), and DLAD (Zeng et al., NeurIPS 2024), we think your argument is valid and will provide readers an intuitive understanding about the detection accuracy. Here is a part of the comparison on ChatGPT generations from Table 6.
>
> ```
> |                             | Best Threshold per Dataset        | Best Threshold across Datasets    | Diff    |
> | Method                      | XSum   | Writing| PubMed | Avg.   | XSum   | Writing| PubMed | Avg.   | Avg.    |
> | Fast-Detect (GPT-J/Neo-2.7) | 0.9600 | 0.9633 | 0.8267 | 0.9167 | 0.9400 | 0.9367 | 0.7567 | 0.8778 | -0.0389 |
> | Fast-Detect (Phi2-2.7B)     | 0.7467 | 0.6967 | 0.7600 | 0.7344 | 0.6767 | 0.6967 | 0.7467 | 0.7067 | -0.0277 |
> | Fast-Detect (Llama3-8B)     | 0.8000 | 0.7700 | 0.7267 | 0.7656 | 0.6900 | 0.7667 | 0.6567 | 0.7044 | -0.0612 |
> | DNA-GPT (GPT-3.5)           | 0.8750 | 0.8592 | 0.8833 | 0.8725 | 0.8108 | 0.5986 | 0.7467 | 0.7187 | -0.1538 |
> | PDE (Fast-Detect, GPT-3.5)  | 0.9633 | 0.9967 | 0.9200 | 0.9600 | 0.9033 | 0.9967 | 0.9167 | 0.9389 | -0.0211 |
> ```
>
> We also assess the ACCs utilizing the optimal threshold across datasets and source models, yielding results that are nearly identical (deviation less than 0.006) to the best threshold across datasets. As the table shows, PDE consistently produces higher ACCs than the baselines. Most crucially, **PDE offers the most stable metric** across datasets and source models, implying that it can identify texts from a variety of domains and source models using a fixed threshold.
>
> Additionally, we have included a section “Appendix E.1 Robustness across Source Models and Domains” and a corresponding summary in Section 3.5 to account for an out-of-domain setup for both source models and text domains. The findings reveal that **PDE maintains the highest ACCs across all the source models and domains** using a threshold from out-of-domain datasets.

---

> ### Author Response · Authors · 2024-11-26
> **A kind enquiry**
>
> Dear Reviewer, we have diligently attended to your previously raised concerns. Would you mind providing us with any additional feedback you might have?
>
> Since our last conversation, we have included additional experimental outcomes we believe sufficiently address your concerns.
>
> Fist, we introduce **Mix3 (450 pairs)**, which is a blend of XSum, Writing, and PubMed, and **Mix6 (900 pairs)**, which is an integration of six languages. These datasets provide diversity in both domain and language, thereby **achieving the required scale and variety**. We have updated the average AUROCs (Avg.) with the AUROCs on the individual dataset, as shown in Table 1, 2, 3, 4, and 5. The adjusted scores **do not change the final conclusion or discussion**.
>
> Next, we have included an **out-of-domain** scenario and provided the **ACCs** for both textual domains and source models, as indicated in Table 8 and 9. **PDE has yielded the highest ACCs in all these settings**.
>
> Last, we have cited "AI-generated text boundary detection with RoFT, Kushnareva et. al, 2024" as pertinent literature. We relocated the appendix section titled "Necessity of PDE" into the main body of the text. For the appendix section "Robustness Under Paraphrasing Attack," we included a summary in Section 3.5, while keeping the detailed content in the appendix due to space constraints.

---

> ### Comment · Reviewer_Gnc2 · 2024-11-29
>
> Sorry for the delay. I checked the updates in your paper, re-read your discussion with me and with other reviewers, and "Reply to Two Common Concerns". After it, I have the following thoughts.
>
> I appreciate the new experiments, new accuracy results, new Appendix E.1 Robustness across Source Models and Domains, the explanations about the number of examples, and answers to my questions that helped me to clarify some parts of your paper. Two of my major concerns were addressed with these additions. Due to this, I update my main score from 3 to 5 and adjust other scores accordingly.
>
> I especially appreciated your answer to the question about curvature. It stimulated me to re-read again the method description in the original paper about DetectGPT and get acquainted with the newer FastDetectGPT paper that introduced some formalisms that you use.
>
> ---
>
> Nevertheless, I would like to point out some other things that I didn't comment on the previous time due to lack of time.
>
> **First:**
>
> > we suspect that your concerns with W1 and W3 might be due to a lack of familiarity with the task of identifying machine-generated text using zero-shot techniques
>
> I found your comment regarding my familiarity with the task dismissive, as it implies a lack of understanding on my part, and it felt more like a personal attack than a constructive critique. I would appreciate it if you could refrain from making comments like that in the future.
>
> **Second:**
>
> The following concern still remains unaddressed:
>
> > AUROC scores in the Tables seem to be very high both for the best baseline and for the best method suggested by the authors. It means that the task statement is too easy. It would be good to add more complicated data, where all the methods will struggle, and the difference between methods will become more clear.
>
> Your answer was the following:
>
> > It is commonplace for zero-shot detection techniques to achieve AUROC scores exceeding 0.90. This can be seen in recent research like DetectGPT by Mitchell et al. (ICML 2023), Fast-DetectGPT by Bao et al. (ICLR 2024), DNA-GPT by Yang et al. (ICLR 2024), and DLAD by Zeng et al. (NeurIPS 2024).
>
> You are right that many papers about artificial text detection use simple datasets that allow us to gain high scores. But if they all do it, it doesn't mean that it is a good thing to do.
>
> I know that the discussion phase is close to its end, so it is too late to ask for a new experiment for this particular paper. However, I recommend you try more complicated datasets in your next research. For a start, I would suggest trying the dataset from the paper _"RAID: A Shared Benchmark for Robust Evaluation of Machine-Generated Text Detectors" by Dugan et al_.
>
> I suggest this not because I want you to cite this paper (I'm not even an author of it, and my scores don't depend on citations anyway) but because I think that researchers in the field should move closer to real-life scenarios, which are usually much more harsh than very simple datasets that you see in most papers on the topic. No detector from the papers you listed will gain 0.9 AUC in real-life scenarios where users of generative models are much more cunning and have much more diverse behavior, than generations in the simple datasets everyone uses. That "RAID" benchmark looks like the most close to some real-life thing. I suppose that datasets from _"SemEval-2024 Task 8: Multidomain, Multimodel and Multilingual Machine-Generated Text Detection" by Wang et.al_ are also "more-or-less okay" option for the future research, yet are still a bit too simple.

---

> ### Author Response · Authors · 2024-11-29
> **Appreciate and reply to further questions regarding W3**
>
> We are grateful for your feedback, particularly for acknowledging our revisions. Thank you for being candid about your thoughts on our potentially inappropriate wording, which will certainly aid us in crafting better responses in the future. We also value your thorough explanation and suggestions regarding experiments on more challenging datasets such as RAID, and we will take this into account in our next revision. However, we argue that our **current experiments have already covered the perspectives provided by RAID** and adequately support our claim as follows.
>
> ---
> > You are right that many papers about artificial text detection use simple datasets that allow us to gain high scores. But if they all do it, it doesn't mean that it is a good thing to do.
>
> **Replay:** We agree that testing with more challenging datasets can highlight the advancement made by the method; there are always more datasets to explore. However, we would like to argue that the **current experiments are sufficient to justify the novelty and contribution** of PDE — specifically, "enabling white-box methods to use proprietary models," which was previously unachievable.
>
> First, more challenging dataset like RAID combines domain coverage, model coverage, and adversarial coverage, except for multilingual coverage. **Our current experiments already address these aspects**: the domain-diverse dataset for domain coverage, the five source models for model coverage, the paraphrasing attack for adversarial coverage, and the language-diverse dataset for multilingual coverage.
>
> Second, the primary aim of this study is to show the feasibility and potential of enabling white-box methods to use proprietary LLMs, not necessarily to outdo all other techniques in every situation. We argue that **this potential is well demonstrated** through its effectiveness across multiple white-box methods and the significant improvements over versions using open-source models.

---

> > ### Author Response · Authors · 2024-12-02
> > **A kind further enquiry**
> >
> > Dear Reviewer, we have thoroughly examined the challenging RAID dataset you suggested. However, we believe that its perspectives are already addressed by our existing experiments and may not add additional value to our study. Could you please review our arguments?

---

### Official Review · Reviewer_32Jz · 2024-11-04

**Soundness:** 2
**Presentation:** 3
**Contribution:** 2
**Rating:** 6
**Confidence:** 4

**Summary:**

In this paper, the authors consider the task of artificial text detection (ATD); one commonly-used approach here is to use a model itself to detect its own output. This scheme is inapplicable to modern proprietary LLMs because they don’t give access to their internal states. To address this issue, authors introduce an extension to existing white-box detection methods that rely on the distribution of model-assigned probabilities to each token. The authors propose to substitute token probability distributions by their density estimations (PDE) based on probabilities of the few most probable tokens and a hypothesis on the shape of the rest of the distribution. They test various candidates for the hypothetical token probability distributions. On example of Fast-DetectGPT authors illustrate the scheme of PDE-approach application to ATD solutions. Authors perform numerous experiments on different types of texts produced by various generating models and show the high performance of their method. They also perform ablation study, investigate the effect of different prompt designs, and explore the effect of paraphrasing adversarial attack.

**Strengths:**

**S1.** The proposed approach shows promising results at increasing the efficiency of previously existing Artificial Text Detection methods against modern generating models.

**S2.** The text of the article is well-written and was easy to follow; extensive mathematical explanations are given.

**S3.** Proposed method is robust against some common adversarial attacks (e.g., automatic paraphrasing of generated texts).

**Weaknesses:**

**W1.** The proposed method still requires the internal information from the source models – model-assigned probabilities of K most probable candidates for each token. Some proprietary models might not provide such information.

**W2.** The novelty of the proposed method compared to `vanilla’ Fast-DetectGPT is somewhat ambiguous. Essentially, this work proposes to compute statistics not from probability distribution over the entire dictionary (as in Fast-DetectGPT), but from probabilities of K most likely tokens and some assumptions on the distribution.

**W3.** Datasets used for experiments are small: 150 pairs of human-written/AI-generated texts each. In particular, there are concerns regarding the generalization capabilities of the proposed. Also, given data is not large enough to properly train the baseline methods (RoBERTa).

**W4.** AUROC is the only quality metric reported. It has a number of limitations, and alone it is not enough to accurately compare different models. This experimental setup does not allow properly accessing the generalization capabilities of the model and discriminating different types of their biases.
For reference, one of the common practices for Zero-shot AI-content detector evaluation is to set a threshold (by fixing FP-rate) on some 'training’ data (i.e., the probability of falsely declaring human-written text as AI-generated) and then report metrics (TPR/TNR/detection accuracy/etc.) on the `evaluation’ data.

**W5.** It is unclear how good this method would generalize across different styles of texts. As it follows from the text, all reported experiments were performed only in “in-domain” setup. Datasets used in this work are very small and style-specific (e.g., scientific-like *PubMed* and free-form fictional *Writing*). It is well known that some classifiers (like RoBERTa-based) excel on `familiar’ data but experience a significant drop in performance when they face texts of styles or genres unseen during training. From the presented results it is impossible to say whether the proposed method will fall into the same pitfall or not.

**Questions:**

**Q1.** DetectGPT (and many other methods) has a known limitation of a drastically high rate of false alarms on texts written by people who are not fluent in English; for example, by undergraduate students preparing for the IELTS exam [1]. Have you considered such scenarios?
This issue becomes more and more crucial as AI-content detection tools become part of automatic anti-plagiarism systems in education in countries where English is taught as second-language.

> [1] W. Liang, M. Yuksekgonul, Y. Mao, E. Wu, and J. Zou. GPT detectors are biased against non-native English writers. In ICLR 2023 Workshop on Trustworthy and Reliable Large-Scale Machine Learning Models.

**Q2.** How big is the difference in performance between your method (PDE + Fast-DetectGPT) and the original Fast-DetectGPT on various data?

**Q3.** Table 3 states that you use RoBERTa-large model as a baseline for experiments on robustness over languages. This is incorrect because RoBERTa was trained only on English-language data. Did you mean XLM-RoBERTa- large, which is actually multilingual, there?

**Q4.** Could you please specify in more detail the experimental setup for assessing the robustness against paraphrasing attacks (Appendix E.1)?

**Q5.** What are the limits for cross-model transferability of your method? (i.e, the proposed method works well when transferred from ChatGPT to GPT-4 and vice versa, but what would happen when it faces older models like GPT-2 or XLNet as source models?)

**Other remarks:**

-	Table 2 caption: instead of “average accuracies”, there should be “average AUROC”. In general, please revise your usage of the phrase “detection accuracy”, because right now it is difficult to understand whether you mean Accuracy (that is not reported by value anywhere) or AUROC.

-	lines 485/511: technically, PHD is a "black-box" method. It does not require the knowledge of what model was used to generate text; it uses inner embeddings from a fixed auxiliary encoder transformer.

-	Code to reproduce the experiments was not provided (this could be done in attachments to submission in or via an anonymous Github repository).

-	Please fix lettercases in paper titles in the references (e.g., “Fast-detectgpt: Efficient zero-shot detection of machine-generated text via conditional probability curvature” should be “Fast-DetectGPT: Efficient Zero-Shot Detection of Machine-Generated Text via Conditional Probability Curvature” in line 552; and so for other articles).

---

> ### Author Response · Authors · 2024-11-18
> **Reply 1 out of 2**
>
> We appreciate your comprehensive feedback! It appears there may be a few fundamental misconceptions about our study. We trust our response will clarify these points for you. Please let us know if you have any further queries.
>
> ---
> > **W1:** The novelty of the proposed method compared to `vanilla’ Fast-DetectGPT is somewhat ambiguous.
>
> **Reply:** The novelty of PDE does not rely on a comparison with vanilla Fast-DetectGPT, as PDE is a versatile method that can be applied to other white-box methods, such as Entropy, Rank, and LogRank. We have shown that PDE is an adaptable method that has facilitated the implementation of white-box methods in proprietary models for the FIRST time, which is the source of its novelty. In section 2, we merely use Fast-DetectGPT as an example to show how to use PDE with white-box methods, and we do not assert that its uniqueness is dependent on this comparison.
>
> ---
> > **W2:** Datasets used for experiments are small: 150 pairs of human-written/AI-generated texts each.
>
> **Reply:** Please refer to the reply to Common Concern 1.
>
> ---
> > **W3:** In particular, there are concerns regarding the generalization capabilities of the proposed. Also, given data is not large enough to properly train the baseline methods (RoBERTa).
>
> **Reply:** We do NOT engage in any model training. The benefit of zero-shot detection methods is their generalization capacity, which is possible because we utilize pre-existing LLMs as the scoring model.  These LLMs have demonstrated strong generalization abilities across various domains and languages. Our tests across different datasets, source models, and languages have verified these capacities. Furthermore, the RoBERTa baseline was trained by OpenAI as described in section 3.1. Again, it's important to note that we do not conduct any model training.
>
> ---
> > **W4:** AUROC is the only quality metric reported. For reference, one of the common practices for Zero-shot AI-content detector evaluation is to set a threshold (by fixing FP-rate) on some 'training’ data (i.e., the probability of falsely declaring human-written text as AI-generated) and then report metrics (TPR/TNR/detection accuracy/etc.) on the `evaluation’ data.
>
> **Reply:** This is NOT true. We present a comparison of TPR against FPR in Figure 6, which allows us to clearly evaluate the performance of different methods under different FPR conditions.
>
> ---
> > **W5:** It is unclear how good this method would generalize across different styles of texts. As it follows from the text, all reported experiments were performed only in “in-domain” setup. Datasets used in this work are very small and style-specific (e.g., scientific-like PubMed and free-form fictional Writing).
>
> **Reply:** This is the first time I've come across the idea of an "in-domain" setup for zero-shot detection techniques. It's a bit confusing since we don't actually train any model with an "in-domain" dataset. Nevertheless, given that different datasets might have varying metric distributions, we also assess these methods by testing them on a combined dataset of XSum, Writing, and PubMed (integrating the three datasets into one) and present the TPR/FPR in Figure 6.

---

> > ### Author Response · Authors · 2024-11-18
> > **Reply 2 out of 2**
> >
> > ---
> > > **Q1:** DetectGPT (and many other methods) has a known limitation of a drastically high rate of false alarms on texts written by people who are not fluent in English.
> >
> > **Reply:** We will cite the work and experiment if time allows.
> >
> > ---
> > > **Q2:** How big is the difference in performance between your method (PDE + Fast-DetectGPT) and the original Fast-DetectGPT on various data?
> >
> > **Reply:** The detailed results are listed in Table 2 (for ChatGPT), Table 4(for GPT-4 and Gemini1.5 Pro), and Table 5 (for Claude-3 Sonnet and Opus). When we average the AUROCs across the source models for each dataset, we obtain
> >
> > ```
> > | Method                     | XSum   | Writing| PubMed |
> > | Fast-Detect (Neo-2.7)      | 0.9368 | 0.9554 | 0.8436 |
> > | PDE (Fast-Detect, GPT-3.5) | 0.9479 | 0.9906 | 0.9504 |
> > |                            |+0.0111 |+0.0352 |+0.1068 |
> > ```
> > The results show that PDE achieves the most significant improvement on PubMed compared to the baseline.
> >
> > ---
> > > **Q3:** Table 3 states that you use RoBERTa-large model as a baseline for experiments on robustness over languages. This is incorrect because RoBERTa was trained only on English-language data.
> >
> > **Reply:** You are right. We have removed this baseline and add other baselines such as Fast-DetectGPT with Phi-2, Qwen2, and Llama3.
> >
> > ---
> > > **Q4:** Could you please specify in more detail the experimental setup for assessing the robustness against paraphrasing attacks (Appendix E.1)?
> >
> > **Reply:** Based on the three datasets XSum, Writing, and PubMed from the main experiments, we paraphrase the machine-generated text using DIPPER with two settings (60L for high lexical diversity and 60O for high order diversity), producing the datasets. We compare the methods on the paraphrased datasets.
> >
> > ---
> > > **Q6:** What are the limits for cross-model transferability of your method? (i.e, the proposed method works well when transferred from ChatGPT to GPT-4 and vice versa, but what would happen when it faces older models like GPT-2 or XLNet as source models?)
> >
> > **Reply:** In this study, we mainly focus on latest proprietary LLMs because the detection of generations from these models are generally hard. In contrast, generations from older or open-source models are much easier for detection, which have been solved well by Fast-DetectGPT with cheaper open LLMs such as Neo-2.7.
> >
> > ---
> > > **Other remarks**
> >
> > **Reply:** We have revised the paper to fix the representation remarks.

---

> > > ### Comment · Reviewer_32Jz · 2024-11-20
> > > **Comments on Reply**
> > >
> > > Thank you for answering my questions and addressing my concerns and for quick updates to the text of the paper; I have also read other reviews and responses to them. However, I still have certain points that I would like to further discuss.
> > >
> > > > This is the first time I've come across the idea of an "in-domain" setup for zero-shot detection techniques. It's a bit confusing since we don't actually train any model with an "in-domain" dataset.
> > >
> > > My choice of words probably was not very accurate. I will try to explain it.
> > >
> > > Your method calculates some metric of a text; if this metric is below a certain threshold, the detector declares it "AI-generated,” otherwise “Human-written”. High AUROC values are the evidence that there is good separation between metric values of human-written and AI-generated texts within each dataset (“in-domain” setup in my words). However, distributions of metrics for Human-written (or AI-generated) texts might be shifted between different domains (text styles/genres/etc.), and your experiment setup does not cover this issue (“out-of-domain” setup in my words).
> > >
> > > For practical applications you need that decision threshold fixed, and my concern was that, depending on domain, the optimal threshold value might fluctuate. Therefore, if you select the threshold on, for example, *XSum*, the performance of this detector on *Writing* dataset might be lower than expected.
> > >
> > > This adds to concerns regarding usage of AUROC as the sole metric that I and other reviewers have.
> > >
> > > Indeed, this issue isn’t that acute for modern practically applicable zero-shot detectors because their decision rules are usually based on much larger and more diverse text collections than those used in your experiments.
> > >
> > > > **Reply:** Based on the three datasets XSum, Writing, and PubMed from the main experiments, we paraphrase the machine-generated text using DIPPER with two settings (60L for high lexical diversity and 60O for high order diversity), producing the datasets. We compare the methods on the paraphrased datasets.
> > >
> > > From what I can see, you report AUROC on paraphrased datasets. I am afraid that the setup of your experiments with paraphrasing does not align with the setup proposed in the original paper. There, a detecting threshold was first fixed on non-paraphrased texts, and after that detector was tested on paraphrased texts (authors report TPR at fixed FPR). The key aim of such experiments is to highlight paraphrasing-caused shifts in distributions of metrics employed by zero-shot detectors that lead to significant performance drop on paraphrased texts.
> > >
> > > **Minor comments:**
> > >
> > > > Furthermore, the RoBERTa baseline was trained by OpenAI as described in section 3.1.
> > >
> > > As far as I know, that RoBERTa-derived model was intended specifically for GPT-2 output detection and has not been updated since then. Such trained detectors often have much lower performance on other models.
> > >
> > > > In reply to **Common Concern 1**.
> > >
> > > I understand your arguments, but I would like to point out that, although those papers use the same datasets, some of them operate on larger samples from them. In Fast-DetectGPT paper, main results were achieved on datasets of 500 pairs (section 3.2 of the corresponding paper). Same goes for DetectGPT (section 5.1, first paragraph).

---

> > > > ### Author Response · Authors · 2024-11-21
> > > > **Reply to further queries**
> > > >
> > > > Thank you very much for your comprehensive clarification regarding the concerns. We are happy to inform you that we have addressed these issues and updated the paper accordingly.
> > > >
> > > > ---
> > > > > However, distributions of metrics for Human-written (or AI-generated) texts might be shifted between different domains (text styles/genres/etc.), and your experiment setup does not cover this issue (“out-of-domain” setup in my words).
> > > >
> > > > **Reply:**  Although the distribution shift was not considered by recent zero-shot detection studies, such as DetectGPT (Mitchell et al., ICML 2023), Fast-DetectGPT (Bao et al., ICLR 2024), DNA-GPT (Yang et al., ICLR 2024), FourierGPT (Yang et al., EMNLP 2024), and DLAD (Zeng et al., NeurIPS 2024), we think the argument is valid and will provide readers an intuitive understanding about the performance in real scenarios.
> > > >
> > > > We have included a section “Appendix E.1 Robustness across Source Models and Domains” and a corresponding summary in Section 3.5 to account for such an out-of-domain setup for both source models and text domains. The findings reveal that **PDE maintains the highest ACCs across all the source models and domains** using a threshold from out-of-domain datasets.
> > > >
> > > > ---
> > > > > I am afraid that the setup of your experiments with paraphrasing does not align with the setup proposed in the original paper. There, a detecting threshold was first fixed on non-paraphrased texts, and after that detector was tested on paraphrased texts (authors report TPR at fixed FPR).
> > > >
> > > > **Reply:** We have updated Table 10 to present TPR at an FPR level of 1%. Interestingly, the trends align with the original AUROCs. Therefore, the **conclusion holds and the discussion remains the same**.
> > > >
> > > > ---
> > > > **Minor comments:**
> > > > > As far as I know, that RoBERTa-derived model was intended specifically for GPT-2 output detection and has not been updated since then. Such trained detectors often have much lower performance on other models.
> > > >
> > > > **Reply:** We agree. It is just one of the baselines that is used widely by recent studies, like DetectGPT (Mitchell et al., ICML 2023) and Fast-DetectGPT (Bao et al., ICLR 2024) and we follow them to include it here as a reference.
> > > >
> > > > ---
> > > > > In Fast-DetectGPT paper, main results were achieved on datasets of 500 pairs (section 3.2 of the corresponding paper). Same goes for DetectGPT (section 5.1, first paragraph).
> > > >
> > > > **Reply:** If you read the papers carefully, you will find that they use **500 pairs for open-source models** but **150 pairs for GPT3, ChatGPT and GPT4**. Actually, we use the datasets from the official release of Fast-DetectGPT directly for ChatGPT and GPT-4, and only generate datasets for Claude3 and Gemini1.5. Other studies using API-based models also follow this setting, like DNA-GPT (Yang et al., ICLR 2024), FourierGPT (Yang et al., EMNLP 2024), and DLAD (Zeng et al., NeurIPS 2024).
> > > >
> > > > The decision is largely driven by **cost considerations**. Open-source models can be run locally on a single GPU, an approach that is both quick and inexpensive. Contrarily, using API-based proprietary models involves significantly higher costs and carbon dioxide emissions. A non-official analysis using publicly available data (https://www.vellum.ai/blog/llama-3-70b-vs-gpt-4-comparison-analysis) estimates that the inference cost of GPT4 is approximately 20 times more than that of Llama3-70B, and about 200 times more than that of Llama3-8B.

---

> ### Author Response · Authors · 2024-11-26
> **A kind enquiry**
>
> Dear Reviewer,
> we have diligently attended to your previously raised concerns. Would you mind providing us with any additional feedback you might have?
>
> Since our previous discussion, we have added more experimental results that we think adequately address your questions.
>
> Initially, we present **Mix3 (450 pairs)**, a combination of XSum, Writing, and PubMed, and **Mix6 (900 pairs)**, a fusion of six languages. This results in datasets that are diverse in terms of domain and language, thereby **fulfilling the desired scale and diversity**. We've updated the average AUROCs (Avg.) with the AUROCs on the single dataset, as evidenced in Table 1, 2, 3, 4, and 5. The revised scores do not alter the conclusion or the discussion.
>
> Secondly, we've incorporated an **out-of-domain** setting for both text domains and source models, as displayed in Table 8 and 9. Across these settings, **PDE has produced the highest ACCs**.
>
> Lastly, we've omitted weaker baselines such as RoBERTa, since our claims are not dependent on comparisons with them. We also discuss the possible implications for non-native writers and reference to the related work in the Broader Impact section 3.5.

---

> > ### Comment · Reviewer_32Jz · 2024-11-26
> >
> > I apologise for the delayed response. Thank you for the detailed answers and for the updated version of the paper. They addressing my concerns regarding the article. I have read the new revision, and I have a small comment about newer additions.
> >
> > C.1) In lines 292-294 you mention that "corresponding LLM texts are generated using the same prefix (30 token)." Could you please clarify - was this setup also used for question answering on *PubMed*, or in that case LLM was prompted with questions only.
> >
> > Overall, I find proposed method quite interesting, and I really appreciate the large number of new experiments and substantial efforts you made to improve this paper.
> > Thus, I will raise my score for this article.

---

> > > ### Author Response · Authors · 2024-11-27
> > > **Appreciate**
> > >
> > > We are so glad to hear that you find this work interesting, which encourages us to pursue further research!
> > >
> > > > C.1) In lines 292-294 you mention that "corresponding LLM texts are generated using the same prefix (30 token)." Could you please clarify - was this setup also used for question answering on PubMed, or in that case LLM was prompted with questions only.
> > >
> > > **Reply:** The LLM was prompted with questions only for PubMed. Thank you for pointing out the inaccurate wording. We have updated it to "corresponding LLM texts are generated using the same prefix (30 tokens for articles or questions for QAs)."

---

### Author Response · Authors · 2024-11-18
**Reply to Two Common Concerns**

We appreciate the reviewer for their comprehensive feedback! We would like to address the two main concerns as follows.

---
> **Common Concern 1:** Datasets used for experiments are small: 150 pairs of human-written/AI-generated texts each.

**Reply:** This dataset setting is **commonly used by recent studies** like DetectGPT (Mitchell et al., ICML 2023), Fast-DetectGPT (Bao et al., ICLR 2024), DNA-GPT (Yang et al., ICLR 2024), FourierGPT (Yang et al., EMNLP 2024), and DLAD (Zeng et al., NeurIPS 2024). We believe that this setting is adequate for assessing a zero-shot detection method (which uses service API) for two main reasons.

First, the total number of samples and API calls for evaluating each method are large in our experiments. We evaluate each method on (300 samples) * (3 datasets) * (5 source models) = **4500 samples** in our main experiments and on (300 samples) * (6 languages) = **1800 samples** in our experiments across languages. **Of ALL the experiments, PDE has the highest AUROC**, significantly outperforming the baselines. Additionally, we run each experiment three times and report the median. The total cost of all the experiments is (4500 samples) * (5 methods) * (3 times) + (1800 samples) * (3 methods) * (3 times) = **83700 API calls**.

Second, compared to other metrics like ACC and Precision/Recall, AUROC is less sensitive to the number of samples because it is an average score (true positive rate) across various false positive rates. Our empirical observations on ChatGPT generations show that the setting yields stable AUROC results (+/-0.0021) when we sample 150 pairs using different random seeds from the entire dataset. The relative order between methods does not change and the **conclusion holds**.

In conclusion, given that re-running the full experiments on more samples could lead to tens of thousands of API calls, resulting in **unnecessary carbon-dioxide emissions and service costs**, we prefer to maintain the current setting for our experiments.

---
> **Common Concern 2:** It is better to use bigger or newer open models (like Phi2-2.7B, Qwen2.5-1.5B or Llama3) than GPT-Neo-2.7B as the baseline.

**Reply:** The official Fast-DetectGPT utilizes GPT-Neo-2.7B, which we have selected as our baseline. Nonetheless, to encompass the latest open LLMs, we have updated our paper to incorporate experiments on Phi2, Qwen2.5, and Llama3, as depicted in Tables 2, 3, 4, and 5. The main results are like:

```
| Method                      | ChatGPT| GPT4   | Sonnet | Opus   | Gemini1.5-Pro| Avg.   |
| Fast-Detect (GPT-J/Neo-2.7) | 0.9615 | 0.9061 | 0.9304 | 0.9519 | 0.8099       | 0.9119 |
| Fast-Detect (Phi2-2.7B)     | 0.7821 | 0.5727 | 0.7151 | 0.7649 | 0.6251       | 0.6920 |
| Fast-Detect (Qwen2.5-7B)    | 0.7937 | 0.7023 | 0.8180 | 0.8545 | 0.7034       | 0.7744 |
| Fast-Detect (Llama3-8B)     | 0.8299 | 0.7554 | 0.8792 | 0.9089 | 0.7979       | 0.8343 |
```

The results indicate that **recent open LLMs underperform compared to Neo-2.7B**. We speculate this could be due to two factors. First, as the study “Smaller language models are better black-box machine-generated text detectors (Mireshghallah et al., 2023)” reports, larger open-source models do not necessarily yield better results. Second, these open-source models may have been partially trained on synthetic data (machine-generated text), which could lead to a distribution more akin to machine text rather than human text. This could disrupt the underlying assumption of Fast-DetectGPT that the scoring model closely aligns with the distribution of human text.

---

> ### Author Response · Authors · 2024-11-22
> **A further revision**
>
> Great thanks to **Reviewer 32Jz** for the mention. We just realize that the "out-of-domain" setting is typically emphasized by recent Binoculars (Hans et al., 2024), which we had missed during our literature review. We have now incorporated this research into our work and have added a section titled “**Appendix E.1 Robustness across Source Models and Domains**”, and a corresponding recap in Section 3.5, to speak to this "out-of-domain" configuration for both text domains and source models. Our results indicate that **PDE maintains the highest ACCs across the source models and domains** using a threshold from out-of-domain datasets.
>
> Additionally, we have also updated the references to the open-source models Phi-2, Qwen2.5, and Llama3, which were not included in the previous version.

---

### Author Response · Authors · 2024-12-02
**Summary of Revisions**

With the generous guidance from our esteemed reviewers, we have made the following major updates:
1. The **title** and **abstract** have been updated to highlight the contribution.
2. Experiments are conducted on the domain-diverse dataset **Mix3 (450 pairs)** and the language-diverse dataset **Mix6 (900 pairs)** to achieve the desired scale and diversity (see Tables 1-5, which address **Common Concern 1**).
3. Experiments are carried out on **Phi2**, **Qwen2.5**, and **Llama3** to include newer and larger open-source models (see Tables 2-5, addressing **Common Concern 2**).
4. **"Out-of-domain" settings** are integrated to evaluate detection accuracy (ACC) using thresholds from out-of-domain datasets across **text domains** and **source models** (see Tables 6, 8, and 9).
5. We present the **TPR at FPT=1%** in the "Robustness Under **Paraphrasing Attack**" section (see Table 10).
6. Additional adjustments have been made for clarity.

---

### Meta-Review · Area_Chair_Br5b · 2024-12-18

**Metareview:**

This work addresses the problem of machine-generated text detection. Existing approaches fall into white-box methods, which require access to the underlying model and are generally more robust, and black-box, which don't require access to the model (meaning they can be applied to proprietary LLms through an API) but are less robust. This paper proposes applying white-box detection methods to closed-source models by estimating the probability distribution in the black-box model using their new method for Probability Distribution Estimation (PDE). The method is general to different forms of white-box detection, and the experiments show using PDE to apply detection methods to closed-source models improves detection performance over their baselines.

Strengths:
- This paper addresses an interesting issue, the application of propriety LLMs to the task of generated-text detection though approximating white-box detection methods (Gnc2), and the results show that the approach demonstrates promise in improving detection accuracy with these models (32Jz, qGG2).
- The paper provides useful ablations in terms of controlling for false positive rates and common adversarial attacks against detection models (32Jz, Gnc2, NxVa)
- The paper is well-written and easy to understand. (32Jz, NxVa)

Weaknesses:
The primary concerns with the paper come from the experimental setup. All of the reviewers note the very small sample size in the evaluation dataset. The paper provides an impressive number of different detection methods, domains, and models in the experiments; however, the implications of these results are limited given the limited sample size and small differences in performance between methods. In addition, one reason the authors list is that they use a small sample size to rerun each experiment and report the median result. However, the variance across these runs - or even if the differences are significant across settings - is not reported.

Another minor concern raised by the reviewers (32Jz, qGG2) is that the application score of the method is limited, as model APIs do not always provide the required information; the authors never discuss this in their response.

Other weaknesses raised by the reviewers were addressed in the author response.

**Additional Comments On Reviewer Discussion:**

See weaknesses; 3 reviewers raised their scores after the discussion period.

---

### Decision · Program_Chairs · 2025-01-22

Accept (Poster)